# Exploring dual-lidar mean and turbulence measurements over Perdigão's complex terrain

Isadora L. Coimbra[1], Jakob Mann[2], José M.L.M. Palma[1], and Vasco T.P. Batista[1]

[1]University of Porto (UPORTO), Faculdade de Engenharia da Universidade do Porto, Rua Dr Roberto Frias s/n, 4200-465 Porto, Portugal

[2]Technical University of Denmark (DTU), Department of Wind and Energy Systems, Frederiksborgsvej 399, 4000 Roskilde, Denmark

**Correspondence:** Isadora Coimbra (isadora.coimbra@fe.up.pt)

**Abstract.** To assess the accuracy of lidars in measuring mean wind speed and turbulence at large distances above the ground as an alternative to tall and expensive meteorological towers, we evaluated three dual-lidar measurements in virtual mast (VM) mode over the complex terrain of the Perdigão-2017 campaign. The VMs were obtained by overlapping two coordinated Range Height Indicator scans, prioritising continuous vertical measurements at multiple heights at the expense of high temporal and spatial synchronisation. Forty-six days of results from three VMs (VM1 on the SW ridge, VM2 in the valley, and VM3 on the NE ridge) were compared against sonic readings (at $80\,\mathrm{m}$ and $100\,\mathrm{m}$ a.g.l.) in terms of $10\,\mathrm{min}$ means and variances, to assess accuracy and the influence of atmospheric stability, vertical velocity, and sampling rate on VM measurements. For mean flow quantities–wind speed ($V_h$), and $u$ and $v$ velocity components–, the $r^2$ values were close to 1 at all VMs, with the lowest equal to 0.948; whereas in the case of turbulence measurements ($u'u'$ and $v'v'$), the lowest was 0.809. Concerning differences between ridge and valley measurements, the average $RMSE$ for the wind variances was $0.295\,\mathrm{m^2\,s^{-2}}$ at the VMs on the ridges. In the valley, under a more complex and turbulent flow, smaller between-beam angle, and lower lidars' synchronisation, VM2 presented the highest variance $RMSE$, $0.600\,\mathrm{m^2\,s^{-2}}$ for $u'u'$. The impact of atmospheric stability on VM measurements also varied by location, especially for the turbulence variables. VM1 and VM3 exhibited better statistical metrics of the mean and turbulent wind under stable conditions, whereas, at VM2, the better results with a stable atmosphere were restricted to the wind variances. We suspect that with a stable and less turbulent atmosphere, the scan synchronisation in the dual-lidar systems had a lower impact on the measurement accuracy. The impact of the zero vertical velocity assumption on dual-lidar retrievals at $80$ and $100\,\mathrm{m}$ a.g.l. in Perdigão was minimal, confirming the validity of the VM results at these heights. Lastly, the VMs' low sampling rate contributed to $33\,\%$ of the overall $RMSE$ for mean quantities and $78\,\%$ for variances at $100\,\mathrm{m}$ a.g.l., under the assumption of a linear influence of the sampling rate on the dual-lidar error. Overall, the VM results showed the ability of this measurement methodology to capture mean and turbulent wind characteristics under different flow conditions and over mountainous terrain. Upon appraisal of the VM accuracy based on sonic anemometer measurements at $80$ and $100\,\mathrm{m}$ a.g.l., we obtained vertical profiles of the wind up to $430\,\mathrm{m}$ a.g.l. To ensure dual-lidar measurement reliability, we recommend a $90°$ angle between beams and a sampling rate of at least $0.05\,\mathrm{Hz}$ for mean and $0.2\,\mathrm{Hz}$ for turbulent flow variables.

# 1 Introduction

To evaluate the wind at higher heights ($> 100\,\mathrm{m}$), measurements from equipment other than anemometric towers are usually employed, as the costs associated with the installation and maintenance of masts scale with height. An alternative to the use of towers at great heights is the wind lidar.

Lidars measure the wind radial velocity up to kilometres of distance, and when employing a single lidar, a homogeneous flow assumption is needed to retrieve the wind vector components. However, under complex wind flow, this may not be a valid assumption, and measurements may present high systematic errors and inaccurate turbulence parameter estimations (Bingöl et al., 2009a, b; Sathe et al., 2011; Pauscher et al., 2016). For turbulence measurements, relevant to wind turbine load calculations, lidar retrievals are susceptible to cross-contamination and volume-averaging errors (Davies et al., 2005).

To reduce wind measurement uncertainty when using a single lidar in complex terrain, some authors have employed wind models to correct for flow distortion in profiling lidar measurements (Pitter et al., 2012; Klaas et al., 2015; Kim and Meissner, 2017). This approach, however, highly depends on the model's configurations and parameterisations (Klaas et al., 2015).

A more reliable solution to a single lidar is using two or more lidars configured to measure the same control volume simultaneously. In the case of three lidars, the three wind vector components can be retrieved from the radial velocities and azimuth and elevation angles (Mann et al., 2008; Sjöholm et al., 2009; Choukulkar et al., 2017). When two lidars are employed, one wind component, such as the vertical velocity, is assumed to be zero, and the other two are estimated. However, a multi-lidar approach implies high equipment costs and difficulties in coordinating and synchronising the lidar beams (Vasiljević et al., 2016). The scan strategy when employing multi-lidars can vary according to the study's objective. Triple-lidar setups were used by Wildmann et al. (2018) to investigate wind turbine wake and by Newman et al. (2016) to assess turbulence measurements. Coplanar Range Height Indicator (RHI) scans were employed to evaluate rotor structures in a valley by Hill et al. (2010), while Calhoun et al. (2006) overlapped RHI scans to retrieve horizontal wind speed profiles in an urban site.

The association of at least two non-collocated lidars measuring multiple heights in a vertical line is called a virtual mast (VM) or virtual tower. Lidars can be configured with stop-and-stare (Damian et al., 2014; Pauscher et al., 2016; Newman et al., 2016; Debnath et al., 2017b; Wittkamp et al., 2021; Liu et al., 2024) or RHI scans (Calhoun et al., 2006; Ng and Hon, 2022; Newsom et al., 2005; Debnath et al., 2017a). Usually, the stop-and-stare has a higher spatial and temporal synchronisation but needs more time to measure at different heights as the equipment accelerates and decelerates from one measurement height to the next. Conversely, continuous vertical measurements of overlapping RHIs cover several heights more quickly, although usually with less accuracy, due to the scans not being entirely temporally and spatially synchronised, which is mainly a problem in an unstable atmosphere (Wittkamp et al., 2021; Choukulkar et al., 2017).

Rothermel et al. (1985) was the first to assess the feasibility of the dual-lidar methodology. Recent studies include experiments in complex terrain (Hill et al., 2010; Cherukuru et al., 2015; Santos et al., 2020; Duscha et al., 2023) and urban environments (Collier et al., 2005; Newsom et al., 2005; Calhoun et al., 2006; Wittkamp et al., 2021). The effect of atmospheric stability on virtual-mast measurements was evaluated by Newman et al. (2016) and Choukulkar et al. (2017) over flat terrains. Under stable atmospheric conditions, Newman et al. (2016) found that $10\,\mathrm{min}$ turbulent fluctuations from a triple-

lidar VM setup aligned closely with Doppler Beam Swinging (DBS) (Strauch et al., 1984) estimations, and diverged in an unstable atmosphere. However, the study did not include sonic measurements at the same height as the virtual mast, later addressed by Choukulkar et al. (2017), who evaluated triple-lidar VM mean measurements against mean sonic observations (at 50–300 m a.g.l., in 50 m increments). The VM results under stable conditions showed smaller errors than in an unstable atmosphere, which was attributed to the higher wind variability in unstable conditions, potentially leading to greater measurement uncertainty.

Despite previous efforts to evaluate multi-lidar measurements, no study has assessed the mean horizontal wind components obtained from two lidar-coordinated RHI scans in a VM mode, with reference sonic anemometer readings, nor investigated second-order wind statistics from dual-lidar RHI retrievals or the influence of atmospheric stability and sampling rate on these data. Therefore, this study explores coordinated dual-lidar RHI measurements, in a VM mode, of the mean and turbulent flow under different wind conditions over Perdigão's complex terrain. The virtual-mast results are evaluated against sonic anemometer data at one or more matching heights in terms of coefficient of determination ($r^2$) and statistical errors ($RMSE$ and $Bias$).

The VM measurements come from the Perdigão-2017 campaign (Fernando et al., 2019), a field experiment that was part of the New European Wind Atlas (NEWA) (Mann et al., 2017). During the campaign, profiler (8) and scanning (18) lidars were deployed (Fernando et al., 2019). The latter operated with different scanning schemes, including RHIs along the ridges, across the ridges (in three transects), and coordinated setups forming dual-lidar measurements. This work focuses on four virtual masts from the experiment, positioned in a transect almost perpendicular to Perdigão's double-ridge and formed by seven WindScanners (WS), not previously analysed. Thus, we needed to assess the measurements' quality compared to reference data, develop a processing and filtering methodology, and explore the capabilities and limitations of these VMs in Perdigão.

The performance of WindScanners in dual and triple measurement setups, staring at a single point, was evaluated by Pauscher et al. (2016), who compared the results with a sonic anemometer (at 188 m a.g.l.) and DBS readings. The study focused on first- and second-order statistics of horizontal wind components measured by three dual-lidar and one triple-lidar configuration. However, the analysis was limited to a single point, correlating the WS measurements without error quantification.

Previous virtual-mast-based studies in Perdigão combined scanning lidars at different positions than those examined here and with a different focus. Bell et al. (2020) evaluated RHI dual- and triple-lidar measurements in 4 locations along the Perdigão valley in a VM mode (from 50–600 m a.g.l.), focusing on the analysis of the valley flow. However, since the lidars were not coordinated, the VM analysis was based on 15 min mean values, and a time window of 60 s between lidar scans was imposed, which restricted the result analysis to only mean quantities. Triple-lidar VM measurements at different distances within the Perdigão's wind turbine wake were investigated by Wildmann et al. (2019), who proposed a new approach to retrieve the turbulence dissipation rate from RHI lidar retrievals.

Beyond the difficulties in multi-lidar measurements, an additional one lies in measuring the complex wind flow above the mountainous terrain of Perdigão. With wind turbines increasingly being placed in complex terrains due to the depletion of flatland and more site constraints, a greater understanding and mapping of the wind in such areas are required. Furthermore,

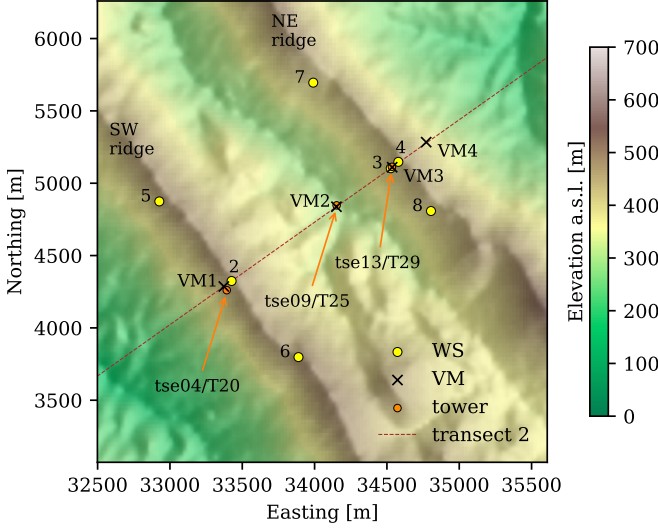

**Figure 1.** Perdigão terrain (Farr et al., 2007) and measuring device locations (ETRS89/PT-TM06).

**Table 1.** Coordinates and elevation of each measurement source.

| Source | Name | Eastings [m] | Northings [m] | Elevation a.s.l. [m] |
|---|---|---|---|---|
| tower | tse04/T20 | 33394.2 | 4258.9 | 473.0 |
| tower | tse09/T25 | 34153.0 | 4844.8 | 305.3 |
| tower | tse13/T29 | 34536.0 | 5111.6 | 452.9 |
| WS | 102 (WS2) | 33426.2 | 4324.1 | 480.3 |
| WS | 103 (WS3) | 34526.4 | 5103.5 | 452.3 |
| WS | 104 (WS4) | 34578.9 | 5147.7 | 454.9 |
| WS | 105 (WS5) | 32926.5 | 4874.3 | 485.9 |
| WS | 106 (WS6) | 33888.7 | 3798.0 | 486.3 |
| WS | 107 (WS7) | 33990.6 | 5695.3 | 437.1 |
| WS | 108 (WS8) | 34804.6 | 4807.9 | 452.8 |

with the growth in height and rotor of modern wind turbines, it is crucial to assess the wind potential and characteristics at greater heights.

## 2 The campaign and equipment

### 2.1 Field campaign

Located in Portugal's mainland, the Perdigão site is characterised by two parallel ridges (SW and NE) with an elevation of about $250\,\mathrm{m}$ above the nearby terrain, separated by $1.4\,\mathrm{km}$, and extending over $4\,\mathrm{km}$, Fig. 1. The SW ridge averages $231.2\,\mathrm{m}$ with a slope of around $33.3°$; the NE ridge is about $217.6\,\mathrm{m}$ with an inclination of $28.5°$; and the valley floor is $41.9\,\mathrm{m}$. The terrain coverage is non-homogeneous, with a mixture of low vegetation and eucalyptus and pine tree patches (Palma et al., 2020).

In the Perdigão-2017 campaign, multiple measuring devices worked simultaneously to obtain a high-resolution dataset from $1^{\mathrm{st}}$ of May until $15^{\mathrm{th}}$ of June 2017. This is called the intensive observational period, IOP, and is the study period of this work. Among the installed equipment, the sensors employed here are those installed in the three $100\,\mathrm{m}$ masts and seven WindScanners operated by the Technical University of Denmark (DTU), Fig. 1.

The wind flow in Perdigão was initially assumed to be two-dimensional, with the predominant wind direction perpendicular to its double ridge (Fernando et al., 2019). However, the measurements revealed Perdigão's intricate wind flow. Despite the uniform perpendicular flow on the synoptic scale, on smaller scales, the wind exhibits two main directions (Fig. 2). In the

valley, the wind direction aligns with the valley (tse09/T25 wind rose), while on the ridges (tse04/T20 and tse13/T29 wind roses), it is perpendicular to the valley.

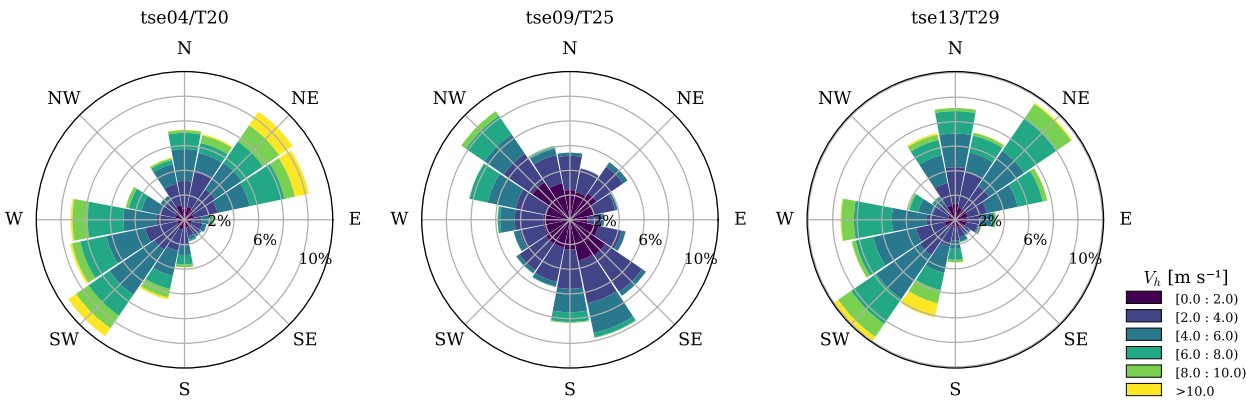

**Figure 2.** Wind roses of the 10 min averaged wind speed and direction from tse04/T20 (SW ridge), tse09/T25 (valley), and tse13/T29 (NE ridge) measurements at 100 m a.g.l. during the intensive observational period.

## 2.2 Towers

The three 100 m towers were located along transect 2 (Menke et al., 2019b), almost perpendicular to the ridges: tse04/T20 on the SW ridge, tse09/T25 in the valley, and tse13/T29 on the NE ridge (Fig. 1 and Table 1). The tower equipment provided wind speed and temperature measurements that were used in this study to evaluate the VM wind speed retrievals and classify the atmospheric stability.

Gill 3D WindMaster Pro sonic anemometers were operated at a frequency of 20 Hz, with sensor heights shown in Table 3 and Fig. 3. NCAR SHT75 temperature/humidity sensors were installed at seven levels: 2 m, 10 m, 20 m, 40 m, 60 m, 80 m, and 100 m a.g.l. The post-processed (quality controlled, tilt-corrected, and in a geographic coordinate system) data from these instruments were downloaded from UCAR/NCAR - Earth Observing Laboratory (2019a). The sonic anemometer data was tilt-corrected using laser survey measurements (Menke and Mann, 2017) to determine the azimuth, pitch, roll, and height of each anemometer, ensuring that the post-processed wind components were represented in geographical coordinates (UCAR/NCAR - Earth Observing Laboratory, 2019b).

### 2.3 WindScanners

Eight WindScanners (WS1–8), four on each ridge and operated by DTU (Vasiljević et al., 2016; Menke et al., 2019a), were employed in the Perdigão-2017 campaign. In terms of settings, the range gate separation (15 m), full-width half maximum of the spatial weighting function (30 m), spatial coverage (from 100 m to 3000 m away from the equipment), elevation step (0.75°), accumulation time (500 ms), and pulse length (200 ns) were identical for all WindScanners. WS1–4, WS6, and WS8

had an elevation range of 36°, while WS5 and WS7 covered an angular range of 18°. The WindScanners 1–4 performed RHI measurements along transect 2, and WindScanners 5–8 operated in a sequence of three scan types, each with a 10 min duration: along the ridge, virtual mast, and transect scans. By crossing WS2–4 RHI measurements with WS5–8 virtual mast scan (also RHI), four virtual masts (VM1–4) were reconstructed with the campaign measurements (Fig. 1 and Table 2).

To guarantee the quality of the WS measurements, before the dual-lidar processing, the WS data were initially filtered out according to the equipment's radial velocity limits ($[-30, 30]$ m s$^{-1}$) and the carrier-to-noise ratio (CNR), where a threshold equal to $-22$ dB (determined from CNR versus radial velocity plots of the multiple WindScanners) was imposed. The WS spectrum data was not stored in the Perdigão campaign; only the processed signal results were. Other filters were employed while processing the VM measurements (Sec. 3.1).

## 3   Virtual mast retrieval

During the Perdigão-2017 experiment, four virtual masts (VM1–4) were configured (Menke et al., 2019a) according to the intersection point between two non-collocated WindScanners (WS$_a$ and WS$_b$), Table 2. Two virtual masts (VM1 and VM3) were located on the top of the SW and NE ridges, another in the valley (VM2), and the last one downhill of the NE ridge (VM4), Fig. 1. VM1–3 were located at distances of 32.4 m, 9.4 m, and 3.3 m, respectively, from tse04/T20, tse09/T25, and tse13/T29 100 m towers to compare VM results with reference equipment at overlapping heights and to map the vertical profile of the wind from 10 m to around 430 m a.g.l.

**Table 2.** Virtual mast coordinates, lidar combinations, and range of elevation angles ($\phi$).

| Virtual | Lidars | | Easting | Northing | Elevation | $\phi_a$ | $\phi_b$ |
|---|---|---|---|---|---|---|---|
| mast | WS$_a$ | WS$_b$ | [m] | [m] | a.s.l [m] | [°] | [°] |
| VM1 | 103 (WS3) | 105 (WS5) | 33372.7 | 4286.2 | 475.0 | 4.1–13.1 | 5.0–21.6 |
| VM2 | 102 (WS2) | 106 (WS6) | 34151.0 | 4837.6 | 304.5 | −4.6–15.6 | −4.2–13.1 |
| VM3 | 102 (WS2) | 107 (WS7) | 34536.4 | 5110.6 | 452.9 | 2.9–12.6 | 8.0–23.0 |
| VM4 | 104 (WS4) | 108 (WS8) | 34771.3 | 5284.0 | 344.7 | −12.1–14.9 | −5.6–7.9 |

### 3.1   Dual-lidar processing and filtering

The processing and filtering of the dual-lidar measurements in Perdigão required the following steps:

Step 1. The radial velocities of WS$_a$ ($v_{ra}$) and WS$_b$ ($v_{rb}$) were interpolated along the beam direction at the VM coordinates (Table 2).

Step 2. The VM heights (Table 3 and Fig. 3) were calculated as the average of the closest WS$_a$ and WS$_b$ measurement heights.

Step 3. Likewise, the VM measurement timestamps were determined by averaging the WS$_a$ and WS$_b$ nearest timestamps.

Step 4. The Cartesian velocity components in the $x$- ($u$) and $y$-directions ($v$) were obtained from the radial velocities ($v_r$) and the azimuth ($\theta$) and elevation ($\phi$) angles of $\mathrm{WS}_a$ and $\mathrm{WS}_b$, assuming that the vertical wind component is zero ($w = 0$), by:

$$
\begin{bmatrix} u \\ v \end{bmatrix} = \begin{bmatrix} \sin(\theta_a)\cos(\phi_a) & \cos(\theta_a)\cos(\phi_a) \\ \sin(\theta_b)\cos(\phi_b) & \cos(\theta_b)\cos(\phi_b) \end{bmatrix}^{-1} \begin{bmatrix} v_{ra} \\ v_{rb} \end{bmatrix}.
\tag{1}
$$

Subsequently, the horizontal wind speed ($V_h$) was calculated.

– Averages and variances of wind velocity components and wind speed were calculated within $10\,\mathrm{min}$ intervals.

Step 5. The VM measurements were filtered in two Steps:

– The first filter aimed to eliminate hard target interference in VM measurements, Sec. 3.1.1.

– The second filter identified the VM minimum quantity of measurements (MQM) within $10\,\mathrm{min}$ intervals, Sec. 3.1.2.

After these processing steps, we ended up with dual-lidar measurements that spanned the atmosphere from $80$ to $305\,\mathrm{m}$ a.g.l. in VM1, $100$ to $430\,\mathrm{m}$ in VM2, $100$ to $330\,\mathrm{m}$ in VM3, and $60$ to $170\,\mathrm{m}$ in VM4; i.e., more than four times the height of conventional tall meteorological towers ($100\,\mathrm{m}$ a.g.l.). We focused our analysis on the measurements from VM1 at $80$ and $100\,\mathrm{m}$, VM2 at $100\,\mathrm{m}$, and VM3 at $100\,\mathrm{m}$, as these were the only measurements obtained at the same height as the sonic anemometer readings, enabling the evaluation of the VM data's reliability.

Upon validating their accuracy, we can use the entire VM dataset in further studies. However, at higher heights, the assumption of zero vertical velocity (Step 4) can reduce the accuracy of the horizontal wind components obtained from dual-lidar measurements, since the increase in beam elevation angles causes the lidar beams to be more aligned with the vertical component of the wind.

### 3.1.1 Hard target filter

Some WS measurements had interference from hard targets, such as terrain, vegetation, and masts, and were, therefore, filtered out. As a result, VM2 and VM3 presented only one measuring height that overlapped with the sonic heights, at around $100\,\mathrm{m}$ a.g.l., while VM1 had two measuring heights that matched the tse04/T20 sonics, at $\sim 80\,\mathrm{m}$ and $\sim 100\,\mathrm{m}$ a.g.l.

### 3.1.2 Minimum quantity of measurement filter

Although the WSs were configured to perform approximately 22 VM scans in each $10\,\mathrm{min}$ measurement period, device restrictions and filtering led to periods with fewer valid scans, as shown in Fig. 4 for VMs' measurements at $100\,\mathrm{m}$ a.g.l. To evaluate the impact of the number of valid scans per $10\,\mathrm{min}$ period on VM measurement accuracy, we computed error indicators for VM1–3 datasets under various filtering thresholds (Table 4). Starting with unfiltered data ($0\,\%$ filter), we defined the minimum number of scans (threshold) required for a $10\,\mathrm{min}$ measurement to be considered valid, progressively increasing the

**Table 3.** Measurement heights (matching heights between the nearby tower and the VM are in bold).

| Name | Height a.g.l. [m] |
|---|---|
| tse04/T20 | 10.3, 19.9, 27.8, 37.0, 57.2, **77.3**, and **97.3** |
| VM1 | **77.9**, **97.0**, 116.2, 135.4, 154.8, 174.3, 193.9, 208.6, 228.5, 248.7, 269.0, 289.7, and 305.0 |
| tse09/T25 | 10.4, 20.5, 30.1, 40.6, 60.2, 80.3, and **97.5** |
| VM2 | **103.9**, 116.8, 129.7, 148.3, 161.2, 174.0, 186.8, 199.7, 218.3, 231.2, 244.1, 257.0, 269.9, 288.8, 301.8, 314.8, 327.9, 341.1, 360.4, 373.7, 387.1, 400.5, 414.1, and 427.7 |
| tse13/T29 | 10.0, 20.0, 30.1, 40.0, 60.2, 80.0, and **97.0** |
| VM3 | **96.0**, 115.7, 130.0, 149.9, 169.8, 184.4, 204.6, 219.3, 239.9, 260.7, 275.8, 297.0, 312.3, and 327.8 |
| VM4 | 60.3, 66.7, 73.0, 79.3, 85.5, 91.8, 98.0, 104.2, 112.0, 118.2, 124.4, 130.6, 136.9, 143.2, 149.5, 155.8, 162.2, 167.0, and 173.4 |

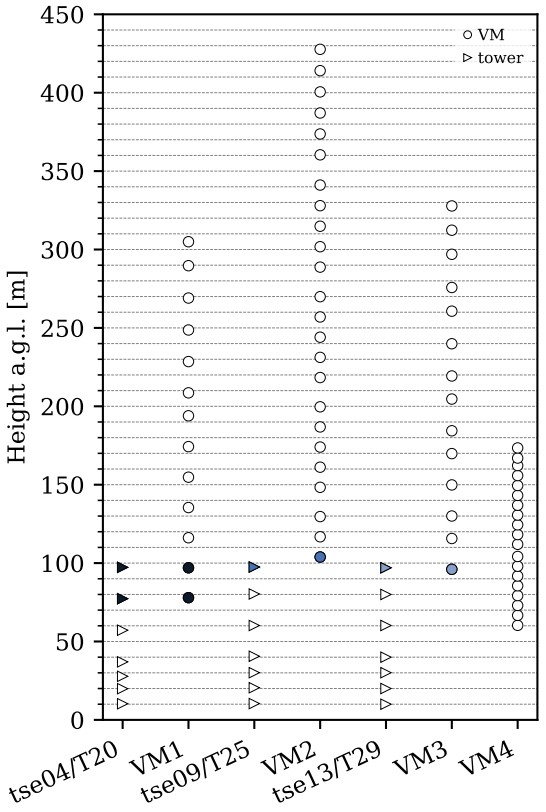

**Figure 3.** Tower and VM heights of wind speed measurements (matching heights in coloured markers: dark blue for tse04/T20 and VM1, medium blue for tse09/T25 and VM2, and light blue for tse13/T29).

filter criteria (as represented by the percentage values in Table 4) up to a $90\,\%$ filter. For example, with the $20\,\%$ filter, a $10\,\mathrm{min}$ measurement was considered valid and included in the analysis if it contained at least $20\,\%$ of the total scan quantity, i.e., four valid scans for a maximum of 22.

The turbulence measurements ($u'u'$ and $v'v'$) were more sensitive to the MQM filter than the mean values ($u$ and $v$), as evidenced mainly by the $RMSE$ (Table 4). Consequently, this metric was chosen as the criterion for identifying the optimal MQM filter value, which retrieves a VM dataset with low errors while avoiding a significant data loss, caused by a too-constrained filter.

    The balance between low $RMSE$ and low data loss occurs when $\Delta RMSE/\Delta N \approx 1$. Here, $\Delta RMSE$ is the percentage
difference in $RMSE$ between any MQM filter above 0 % and the raw data (0 % filter), and $\Delta N$ is the percentage difference in the number of samples between the two datasets. By averaging $\Delta RMSE/\Delta N$ across all VMs, we determined that the optimal MQM filter is $50\,\%$ for the mean and $80\,\%$ for the turbulence VM measurements. Applying a filter higher than $50\,\%$ ($80\,\%$)

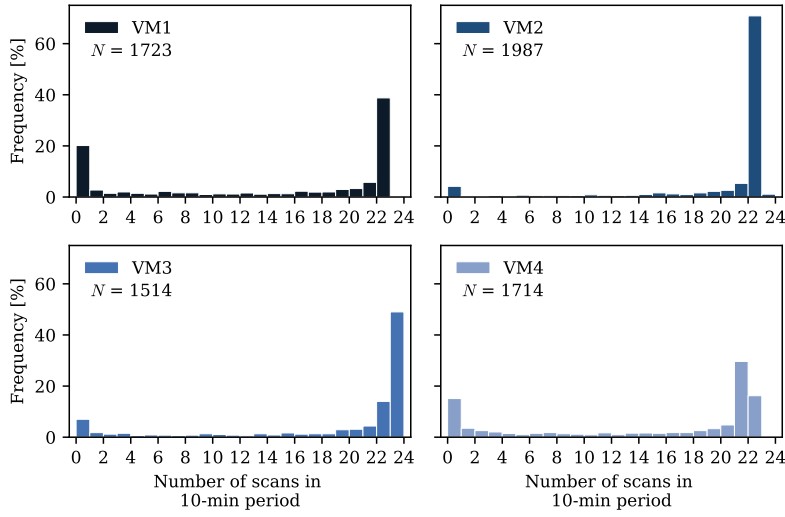

**Figure 4.** Histogram of the number of valid scans in 10 min periods for all virtual masts at ∼100 m a.g.l., before the MQM filter. $N$ represents the total number of valid 10 min measurements at ∼100 m a.g.l. during the IOP, before the MQM filter.

can reduce the dataset size to a point where the remaining data becomes less representative of the mean (turbulent) wind flow. Therefore, subsequent mean and turbulence results will be presented using 50 % and 80 % MQM filters.

Since VM4 is the only virtual mast with no reference measurement nearby, the filtering procedure determined through the VM1–3 analysis was replicated at VM4.

### 3.2 Dual-lidar measurement constraints and error sources

As two simultaneous WSs are required to produce a VM measurement, the VM is constrained by the availability of both WindScanners. WS2–4 (WS$_a$ in Table 2) continuously performed RHI scans, while WS5–8 (WS$_b$ in Table 2) only did the intercepting RHI scan twice per hour. Thus, the VM measurements occurred twice per hour within 10 minutes. During the 10 min period, each WS performed a maximum of 22 or 23 scans (Fig. 4); i.e., a maximum sampling rate of 0.038 Hz (23/600 Hz), approximately 500 times lower than the sonic anemometer frequency (equal to 20 Hz).

Another constraint was the dependence of VM data availability on concurrent measurements from both WindScanners, which, at specific periods, depicted limited data due to equipment downtime or filtering (low CNR, hard targets, and MQM filter). The data availability for each VM at 100 m a.g.l. during the IOP is detailed in Table 5. For mean wind components, the average data availability for all heights was 46.2 % for VM1, 76.3 % for VM2, 54.1 % for VM3, and 56.9 % for VM4. For $u'u'$ and $v'v'$, on the other hand, availability was 37.5 %, 69.8 %, 47.8 %, and 49.2 % for VM1–4.

**Table 4.** Errors between VMs and towers according to the minimum quantity of measurements (MQM) in 10 min periods for $u$, $v$, $u'u'$, and $v'v'$.

| MQM filter | VM1 80 m | | | VM1 100 m | | | VM2 100 m | | | VM3 100 m | | |
|---|---|---|---|---|---|---|---|---|---|---|---|---|
| | $r^2$ | $RMSE$ | $Bias$ | $r^2$ | $RMSE$ | $Bias$ | $r^2$ | $RMSE$ | $Bias$ | $r^2$ | $RMSE$ | $Bias$ |
| $u$ | | | | | | | | | | | | |
| 0 % | 0.993 | 0.496 | 0.366 | 0.992 | 0.536 | 0.377 | 0.982 | 0.559 | 0.488 | 0.993 | 0.654 | 0.582 |
| 20 % | 0.997 | 0.419 | 0.365 | 0.997 | 0.434 | 0.380 | 0.985 | 0.543 | 0.484 | 0.995 | 0.631 | 0.573 |
| 40 % | 0.998 | 0.404 | 0.360 | 0.998 | 0.424 | 0.381 | 0.987 | 0.541 | 0.487 | 0.995 | 0.629 | 0.575 |
| 60 % | 0.998 | 0.395 | 0.354 | 0.998 | 0.416 | 0.377 | 0.987 | 0.540 | 0.489 | 0.996 | 0.623 | 0.575 |
| 80 % | 0.998 | 0.387 | 0.352 | 0.998 | 0.411 | 0.377 | 0.987 | 0.539 | 0.490 | 0.996 | 0.618 | 0.572 |
| $v$ | | | | | | | | | | | | |
| 0 % | 0.986 | 0.524 | $-0.292$ | 0.981 | 0.598 | $-0.292$ | 0.983 | 0.330 | $-0.159$ | 0.995 | 0.369 | $-0.241$ |
| 20 % | 0.993 | 0.421 | $-0.291$ | 0.994 | 0.421 | $-0.307$ | 0.986 | 0.309 | $-0.154$ | 0.997 | 0.333 | $-0.240$ |
| 40 % | 0.995 | 0.385 | $-0.293$ | 0.995 | 0.405 | $-0.311$ | 0.986 | 0.305 | $-0.154$ | 0.997 | 0.320 | $-0.238$ |
| 60 % | 0.996 | 0.370 | $-0.280$ | 0.995 | 0.402 | $-0.313$ | 0.987 | 0.299 | $-0.154$ | 0.998 | 0.312 | $-0.238$ |
| 80 % | 0.996 | 0.355 | $-0.273$ | 0.996 | 0.389 | $-0.307$ | 0.987 | 0.298 | $-0.155$ | 0.998 | 0.306 | $-0.237$ |
| $u'u'$ | | | | | | | | | | | | |
| 0 % | 0.645 | 0.422 | $-0.136$ | 0.756 | 0.319 | $-0.104$ | 0.797 | 0.675 | 0.132 | 0.839 | 0.443 | $-0.165$ |
| 20 % | 0.790 | 0.311 | $-0.127$ | 0.797 | 0.288 | $-0.089$ | 0.818 | 0.632 | 0.135 | 0.859 | 0.429 | $-0.163$ |
| 40 % | 0.845 | 0.259 | $-0.110$ | 0.832 | 0.254 | $-0.083$ | 0.831 | 0.610 | 0.138 | 0.872 | 0.412 | $-0.156$ |
| 60 % | 0.861 | 0.247 | $-0.106$ | 0.849 | 0.241 | $-0.081$ | 0.837 | 0.596 | 0.134 | 0.894 | 0.368 | $-0.147$ |
| 80 % | 0.885 | 0.217 | $-0.094$ | 0.878 | 0.213 | $-0.084$ | 0.833 | 0.600 | 0.131 | 0.895 | 0.357 | $-0.143$ |
| $v'v'$ | | | | | | | | | | | | |
| 0 % | 0.656 | 0.520 | $-0.161$ | 0.686 | 0.477 | $-0.132$ | 0.884 | 0.406 | $-0.022$ | 0.842 | 0.441 | $-0.101$ |
| 20 % | 0.743 | 0.443 | $-0.136$ | 0.744 | 0.424 | $-0.117$ | 0.893 | 0.388 | $-0.021$ | 0.870 | 0.401 | $-0.090$ |
| 40 % | 0.793 | 0.370 | $-0.114$ | 0.799 | 0.369 | $-0.105$ | 0.908 | 0.357 | $-0.020$ | 0.879 | 0.387 | $-0.087$ |
| 60 % | 0.801 | 0.361 | $-0.113$ | 0.812 | 0.363 | $-0.105$ | 0.911 | 0.354 | $-0.017$ | 0.894 | 0.356 | $-0.086$ |
| 80 % | 0.809 | 0.325 | $-0.107$ | 0.818 | 0.330 | $-0.103$ | 0.913 | 0.350 | $-0.023$ | 0.905 | 0.329 | $-0.081$ |

The $RMSE$ and $Bias$ units are $[\mathrm{m\,s^{-1}}]$ for $u$ and $v$ variables, while for $u'u'$ and $v'v'$ are $[\mathrm{m^2\,s^{-2}}]$. $r^2$ is unitless.

The interception angle ($\Delta\chi$) between lidars' beams (Table 6), with directions $\hat{\mathbf{r}}_a$ and $\hat{\mathbf{r}}_b$, influences the accuracy of VM results. This is because the dual-lidar error of a retrieved wind field component ($\sigma_{\mathrm{DD}}(u_j)$) is (Stawiarski et al., 2013):

$$\sigma_{\mathrm{DD}}(u_j) = \left[\frac{\sin^2(\alpha_j + \Delta\chi/2) + \sin^2(\alpha_j - \Delta\chi/2)}{\sin^2\Delta\chi}\right]^{1/2}\sigma_{v_r}, \tag{2}$$

**Table 5.** Data availability of the VM measurements at $100\,\mathrm{m}$ a.g.l. during the IOP.

| Virtual | Mean speed | Turbulence |
|---------|-----------|-----------|
| VM1 | 48.6 % (1073 periods of 10 min) | 39.7 % (876 periods of 10 min) |
| VM2 | 80.8 % (1784 periods of 10 min) | 73.9 % (1632 periods of 10 min) |
| VM3 | 56.0 % (1236 periods of 10 min) | 50.4 % (1112 periods of 10 min) |
| VM4 | 52.4 % (1158 periods of 10 min) | 43.9 % (969 periods of 10 min) |

where $\left[\frac{\sin^2(\alpha_j+\Delta\chi/2)+\sin^2(\alpha_j-\Delta\chi/2)}{\sin^2\Delta\chi}\right]^{1/2}$ is the error prefactor, $\alpha_j$ is the angle between the direction of the wind field component ($\hat{\mathbf{e}}_{\mathbf{j}}$) and the mean lidar direction ($\hat{\mathbf{r}}_{\mathbf{m}} = (\hat{\mathbf{r}}_{\mathbf{a}}+\hat{\mathbf{r}}_{\mathbf{b}})/2$), and $\sigma_{v_r}$ is the radial velocity error, assuming that is identical in both lidars ($\sigma_{v_r} = \sigma_{v_r}^a = \sigma_{v_r}^b$). While the radial velocity error depends on several factors, such as the specific lidar, atmospheric backscatter, distance from the instrument, focus position, and instrument temperature, we assume it to be identical in both lidars because the angle between the beams is a more significant contributor to the dual-lidar error.

**Table 6.** Average angle between lidars' beams ($\Delta\chi$) and prefactors of the dual-lidar propagation error for the horizontal velocity components ($u$ and $v$).

| Virtual mast | $\Delta\chi$ [°] | Prefactors $u$ | $v$ |
|--------------|------------------|----------------|-----|
| VM1 | 89.5 | 1.0 | 1.0 |
| VM2 | 40.2 | 1.8 | 1.3 |
| VM3 | 80.3 | 0.9 | 1.1 |
| VM4 | 58.4 | 1.4 | 1.0 |

The prefactor is directly influenced by the between-beam angle and the direction of the wind component, namely $u$ and $v$, and indirectly by the VM height (Fig. 5), as $\Delta\chi$ varies with the beams' elevation angles. Ideally, the angle between the beams would be close to $90°$, which results in prefactors equal to 1, regardless of the wind component direction. At Perdigão's four virtual masts, only VM1 and VM3 had $\Delta\chi$ close to the optimal angle ($\sim 89.5°$ and $\sim 80.3°$), while the angles at VM2 and VM4 were $40.2°$ and $58.4°$, on average (Table 6). This means that the prefactors and the propagation of the radial velocity error at VM2 and VM4 are greater than at VM1 and VM3.

When retrieving the $u$ velocity, the dual-lidar propagation error is about 1.0, 1.8, 0.9, and 1.4 times the error of the radial velocity for VM1–4, respectively (Table 6). For the $v$ velocity, the prefactors are around 1.0, 1.3, 1.1, and 1.0 for VM1–4. On the other hand, the dual-lidar error of the horizontal wind speed is a combination of the $\sigma_{\mathrm{DD}}(u)$, $\sigma_{\mathrm{DD}}(v)$, and wind speed components:

$$\sigma_{\mathrm{DD}}(V_h) = \left[\left(\frac{u}{\sqrt{u^2+v^2}}\sigma_{\mathrm{DD}}(u)\right)^2 + \left(\frac{v}{\sqrt{u^2+v^2}}\sigma_{\mathrm{DD}}(v)\right)^2\right]^{1/2}, \tag{3}$$

assuming that the errors in $u$ and $v$ are not correlated.

With regard to height variation (Fig. 5), the prefactors varied little and generally showed higher values with increasing height, except for the $v$-wind component measured by VM1.

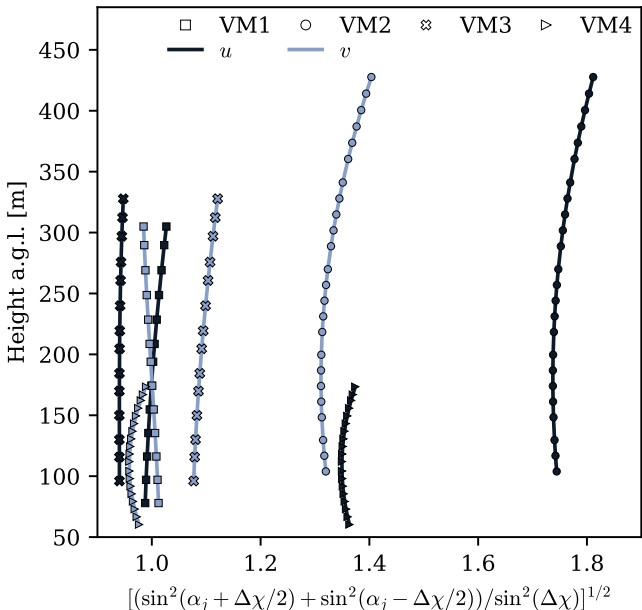

**Figure 5.** Dual-lidar error prefactor ($[(\sin^2(\alpha_j + \Delta\chi/2) + \sin^2(\alpha_j - \Delta\chi/2))/\sin^2(\Delta\chi)]^{1/2}$) of a retrieved wind field component as a function of the beam height for VM1–4.

Another source of error when combining radial velocities from different lidars can arise when there is a mismatch in their range gate heights (Stawiarski et al., 2013). Such mismatch can cause the lidars to measure different wind structures, mainly under high vertical wind shear conditions. For the Perdigão-2017 campaign, the height difference of the central of the control volume, after the radial interpolation, varied for each height and virtual mast. At VM1–4, the displacements went up to $4.4\,\mathrm{m}$, $6.8\,\mathrm{m}$, $8.7\,\mathrm{m}$, and $1.6\,\mathrm{m}$, respectively. However, given that the spatial resolution of the WindScanners was approximately $30\,\mathrm{m}$, this mismatch is not expected to impact the virtual mast results substantially.

In addition, the lidars' scans were not fully synchronised in time (Fig. 6). This means that measurements from $\mathrm{WS}_a$ and $\mathrm{WS}_b$ occurred at slightly different times, which can lead to time-average errors in the dual-lidar measurements (Stawiarski et al., 2013) due to the stationary atmospheric assumption (Choukulkar et al., 2017). At VM1, the predominant time differences between $\mathrm{WS}_a$ and $\mathrm{WS}_b$ ranged from 0 to $2\,\mathrm{s}$, accounting for $53.7\,\%$ of all VM1 measurements. At VM2, $\mathrm{WS}_b$ consistently recorded measurements later than $\mathrm{WS}_a$, leading to time lags of $8$–$10\,\mathrm{s}$ in $69.8\,\%$ of VM2's measurements. For VM3, $51.1\,\%$ of the measurements depicted a time difference between $3\,\mathrm{s}$ and $5\,\mathrm{s}$. Meanwhile, at VM4, the time difference for $62.8\,\%$ of the measurements fell in the $[1\,\mathrm{s}, 3\,\mathrm{s})$ interval. While these desynchronisations may impact the retrieval of turbulent variables, their influence is expected to be insignificant for mean quantities.

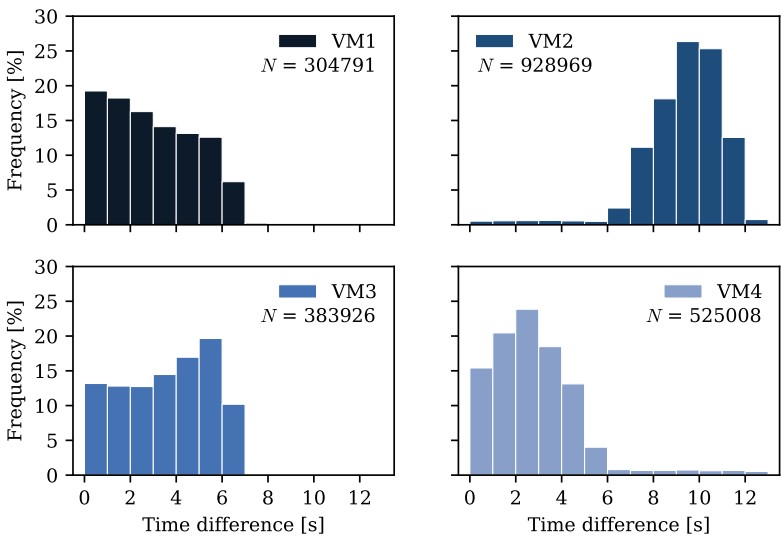

**Figure 6.** Time difference histogram of the mean flow measurements at all heights between the lidars constituting the virtual masts. $N$ represents the total number of valid $10\,\mathrm{min}$ measurements at all heights during the IOP.

Lastly, the horizontal position of each VM differed from the corresponding tower locations. This can affect the VM results when nearby tower measurements are used as a reference due to the underlying assumption of a spatially homogeneous atmosphere. This is most pronounced for VM1, located $32.4\,\mathrm{m}$ apart from tse04/T20. Meanwhile, VM2 was $9.4\,\mathrm{m}$ from tse09/T25, and VM3, $3.3\,\mathrm{m}$ from tse13/T29.

## 4 Results and discussion

This section compares virtual mast and sonic measurements and how atmospheric stability, vertical velocity, and sampling rate influence the VM wind velocity and turbulence retrievals. The analyses are based on $10\,\mathrm{min}$ averages of the horizontal wind speed ($V_h$) and its components ($u$ and $v$), as well as their variances ($u'u'$ and $v'v'$). The virtual mast and sonic comparisons also cover radial velocity means ($v_r$) and variances ($v_r'v_r'$). All results are in local time, equal to UTC + $1\,\mathrm{h}$ in the summer period, and in the ETRS89/PT-TM06 coordinate system.

### 4.1 Virtual mast and sonic comparisons

Virtual mast and tower measurements were compared at their closest heights, with no vertical interpolation: VM1 at $77.9\,\mathrm{m}$ and $97\,\mathrm{m}$ with tse04/T20 at $77.3\,\mathrm{m}$ and $97.3\,\mathrm{m}$; VM2 at $103.9\,\mathrm{m}$ with tse09/T25 at $97.5\,\mathrm{m}$; and VM3 at $96.0\,\mathrm{m}$ with tse13/T29 at $97.0\,\mathrm{m}$. For simplification, the comparison heights were rounded to $80\,\mathrm{m}$ and $100\,\mathrm{m}$.

As a first analysis, $v_{r_a}$ and $v_{r_b}$ from the WindScanners of VM1–3 were compared against sonic measurements projected in the laser beam direction, to assess the measurements of each WS equipment without introducing uncertainties related to the dual-lidar methodology (Sec. 3).

Care must be taken when comparing VM results in the valley (VM2) with those on the ridges (VM1 and VM3), since the flows are intrinsically different at the comparison heights (80 and 100 m a.g.l.). In the valley, the main wind direction is along the valley, whereas on the ridges is cross-valley; the wind speeds are lower (Fig. 2); and the turbulence intensity is 2.7 times higher than on the ridges.

### 4.1.1 Mean flow measurements

In the comparison between VM and sonic $v_r$ (Table 7), the fit of the linear regressions for all WindScanners was almost perfect, with $r^2$ values close to 1. The lowest $r^2$ was equal to 0.989 (WS6 at VM2 100 m). In the linear regression equation ($y = mx + b$), despite the coefficients ($m$) being approximately one, the constants ($b$), determined by where the line intercepts the $y$-axis, assumed positive (WS5, WS2, and WS7) and negative (WS3 and WS6) values according to the WS, meaning an overall overestimation and underestimation of $v_r$. In addition, $b$ higher than $0.4 \, \text{m s}^{-1}$ were observed in WS5 ($0.414 \, \text{m s}^{-1}$ at 80 m and $0.445 \, \text{m s}^{-1}$ at 100 m a.g.l.) and WS7 ($0.492 \, \text{m s}^{-1}$ at 100 m a.g.l.). These WindScanners also showed higher $RMSE$ and $Bias$ errors in their radial velocities at 100 m, $0.509 \, \text{m s}^{-1}$ and $0.436 \, \text{m s}^{-1}$ in WS5 and $0.586 \, \text{m s}^{-1}$ and $0.523 \, \text{m s}^{-1}$ in WS7.

When WS5 and WS7 form VM1 and VM3, their beams align with the direction of the ridges (Fig. 1) and, at the top of the hills, the main wind directions are perpendicular to the ridge's orientation (Fig. 2). Thus, due to a lidar's inherent limitation to directly measure the wind component perpendicular to its beam orientation, WS5 and WS7 setups contribute to their wind speed measurement errors.

For the horizontal wind speed ($V_h$) and $u$ and $v$ wind components obtained from the dual-lidars, besides the beam orientation of each WS regarding the position of the wind, the intersection angle between the two beams is also important (Table 6). At VM1 and VM3, $\Delta\chi$ was close to $90°$, the optimal angle to retrieve $u$ and $v$; whereas, at VM2, the angle was about $40°$, yielding higher dual-lidar propagation error in the $u$ and $v$ components, with mean prefactors equal to 1.8 and 1.3 (Table 6 and Fig. 5).

The coefficients of determination were close to 1 for the mean wind variables at all virtual masts (Table 7 and Fig. 7), with the lowest values equal to 0.987 for $u$ and $v$ and 0.948 for $V_h$ at VM2. The lower $r^2$ values at VM2 are attributed to the smaller angle between WS2 and WS6 beams and to the turbulent flow in the valley, which may require a greater VM sampling rate than 0.038 Hz. The highest errors, however, occurred at VM3 for $u$ ($0.626 \, \text{m s}^{-1}$ $RMSE$ and $0.575 \, \text{m s}^{-1}$ $Bias$) and at VM1 for $v$ ($0.401 \, \text{m s}^{-1}$ $RMSE$ and $-0.310 \, \text{m s}^{-1}$ $Bias$); while for the horizontal wind speed, VM3 obtained the highest $RMSE$, equal to $0.463 \, \text{m s}^{-1}$, and VM2 the highest $Bias$, $0.188 \, \text{m s}^{-1}$. Additionally, all VM results overestimated the anemometer readings of the mean east-west wind component and $V_h$ (positive $Bias$), and underestimated the north-south wind component (negative $Bias$).

The average magnitude of the VM error ($RMSE$) did not follow the trend observed in the dual-lidar propagation errors. Contrary to the prefactor values (Table 6), VM2's $u$ variable did not show the highest $RMSE$ value among the VMs, and the

**Table 7.** Statistical parameters from VM and tower comparisons for mean and variance variables.

| Height a.g.l. [m] | Metric | Mean speed | | | | | Turbulence | | | |
|---|---|---|---|---|---|---|---|---|---|---|
| | | $v_{ra}$ | $v_{rb}$ | $u$ | $v$ | $V_h$ | $v'_{ra}v'_{ra}$ | $v'_{rb}v'_{rb}$ | $u'u'$ | $v'v'$ |
| **VM1 (SW ridge):** | | WS3 | WS5 | | | | WS3 | WS5 | | |
| 80 | $m$ | 1.016 | 0.992 | 0.992 | 1.018 | 1.007 | 0.861 | 0.847 | 0.914 | 0.799 |
| | $b$ | −0.140 | 0.414 | 0.364 | −0.283 | 0.080 | −0.026 | −0.013 | −0.049 | 0.015 |
| | $r^2$ | 0.999 | 0.990 | 0.998 | 0.995 | 0.981 | 0.821 | 0.875 | 0.885 | 0.809 |
| | $RMSE$ | 0.230 | 0.486 | 0.398 | 0.375 | 0.342 | 0.233 | 0.288 | 0.217 | 0.325 |
| | $Bias$ | −0.151 | 0.409 | 0.356 | −0.285 | 0.112 | −0.096 | −0.109 | −0.094 | −0.107 |
| 100 | $m$ | 1.022 | 0.985 | 1.002 | 1.010 | 1.007 | 0.861 | 0.817 | 0.894 | 0.773 |
| | $b$ | −0.150 | 0.445 | 0.377 | −0.306 | 0.071 | −0.024 | 0.015 | −0.029 | 0.034 |
| | $r^2$ | 0.999 | 0.991 | 0.998 | 0.995 | 0.982 | 0.828 | 0.867 | 0.878 | 0.818 |
| | $RMSE$ | 0.253 | 0.509 | 0.419 | 0.401 | 0.356 | 0.238 | 0.278 | 0.213 | 0.330 |
| | $Bias$ | −0.153 | 0.436 | 0.378 | −0.310 | 0.105 | −0.093 | −0.097 | −0.084 | −0.103 |
| **VM2 (valley):** | | WS2 | WS6 | | | | WS2 | WS6 | | |
| 100 | $m$ | 1.055 | 1.044 | 1.023 | 1.036 | 1.047 | 0.849 | 0.888 | 1.269 | 1.031 |
| | $b$ | 0.285 | −0.039 | 0.480 | −0.153 | 0.078 | −0.061 | −0.048 | −0.130 | −0.053 |
| | $r^2$ | 0.993 | 0.989 | 0.987 | 0.987 | 0.948 | 0.936 | 0.935 | 0.833 | 0.913 |
| | $RMSE$ | 0.345 | 0.227 | 0.541 | 0.300 | 0.443 | 0.372 | 0.341 | 0.600 | 0.350 |
| | $Bias$ | 0.303 | −0.036 | 0.489 | −0.155 | 0.188 | −0.205 | −0.158 | 0.131 | −0.023 |
| **VM3 (NE ridge):** | | WS2 | WS7 | | | | WS2 | WS7 | | |
| 100 | $m$ | 0.993 | 1.037 | 0.995 | 1.023 | 1.027 | 0.853 | 0.952 | 0.815 | 0.977 |
| | $b$ | 0.315 | 0.492 | 0.577 | −0.221 | 0.026 | −0.022 | −0.058 | −0.010 | −0.063 |
| | $r^2$ | 0.999 | 0.992 | 0.995 | 0.997 | 0.965 | 0.956 | 0.888 | 0.895 | 0.905 |
| | $RMSE$ | 0.346 | 0.586 | 0.626 | 0.317 | 0.463 | 0.261 | 0.343 | 0.357 | 0.329 |
| | $Bias$ | 0.315 | 0.523 | 0.575 | −0.236 | 0.152 | −0.128 | −0.095 | −0.143 | −0.081 |

The units of $b$, $RMSE$, and $Bias$ are $[\mathrm{m\,s^{-1}}]$ for mean variables and $[\mathrm{m^2\,s^{-2}}]$ for variances. $m$ and $r^2$ are unitless. $m$ is the coefficient, and $b$ is the constant of the linear regression equation ($y = mx + b$).

$x$-wind component in VM3 did not exhibit the lowest, indicating that factors beyond the error coefficient influenced the VMs' $RMSE$.

The $V_h$ errors of the VMs generally fell within the range of those for the $u$ and $v$ components. The $r^2$, on the other hand, showed lower values (0.969 on average) than for $u$ and $v$ (0.994 on average).

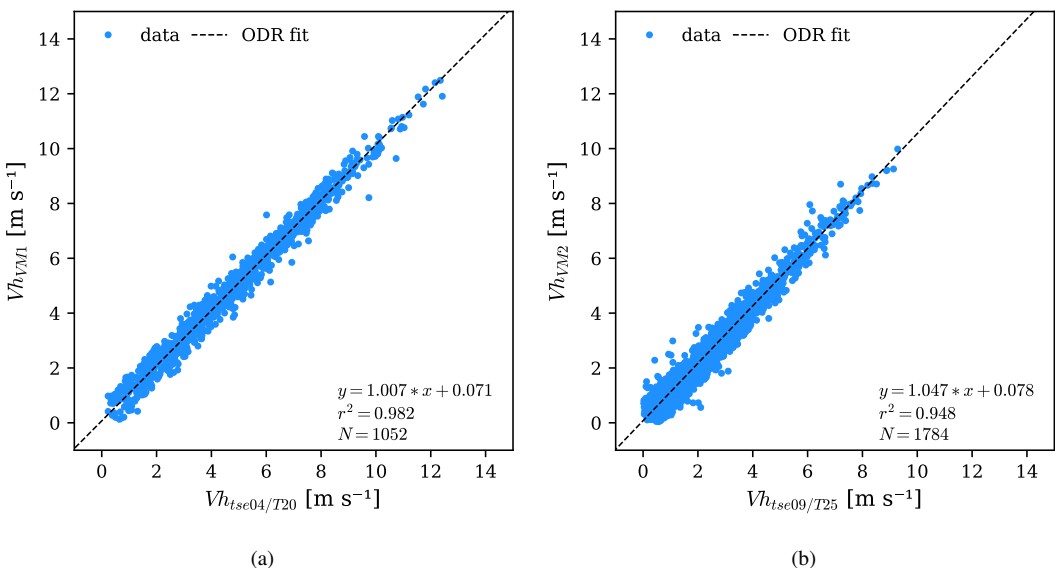

**Figure 7.** Mean flow measurements of virtual masts against sonic anemometer data: (a) VM1 and tse04/T20 $V_h$ at 100 m a.g.l. and (b) VM2 and tse09/T25 $V_h$ at 100 m a.g.l.

Compared to Pauscher et al. (2016), the horizontal wind speed results of Perdigão's VMs showed lower $r^2$ values against reference sonic anemometer measurements, $\sim 3\%$ lower on average. The difference between both results is due to the scanning mode and the underlying assumptions in each scan. Pauscher et al. (2016) employed a staring configuration, recording data at 0.5 Hz, whereas, in our analysis, the virtual mast measurements were formed by combining two RHI scans with a maximum

sampling rate of 0.038 Hz. In the latter, the lidar beams were constantly moving and not perfectly synchronised in time and space, resulting in a lower measurement frequency and forcing a greater flow homogeneity assumption compared to the staring approach.

### 4.1.2 Turbulence measurements

For the radial velocity variances ($v_{r_a}'v_{r_a}'$ and $v_{r_b}'v_{r_b}'$), the $r^2$ values were consistently lower than for the mean radial velocities

($v_{r_a}$ and $v_{r_b}$), going from 0.994 in the means to 0.888 in the variances, on average (Table 7). The lowest coefficient of determination for $v_r'v_r'$ between lidar and sonic measurements was 0.821 at WS3 in VM1 80 m, whereas the highest was 0.956 at WS2 in VM3 100 m.

The radial velocity variance errors averaged $0.294\,\mathrm{m^2\,s^{-2}}$ for $RMSE$ and $-0.123\,\mathrm{m^2\,s^{-2}}$ for $Bias$ on the ridges. In the valley, under a more turbulent flow and with a low measurement rate, the average errors for $v_r'v_r'$ were higher than those on

the ridges, with an $RMSE$ of $0.357\,\mathrm{m^2\,s^{-2}}$ and $Bias$ of $-0.182\,\mathrm{m^2\,s^{-2}}$. However, independent of the measurement location, all WindScanners underestimated the turbulence measurements (negative $Bias$).

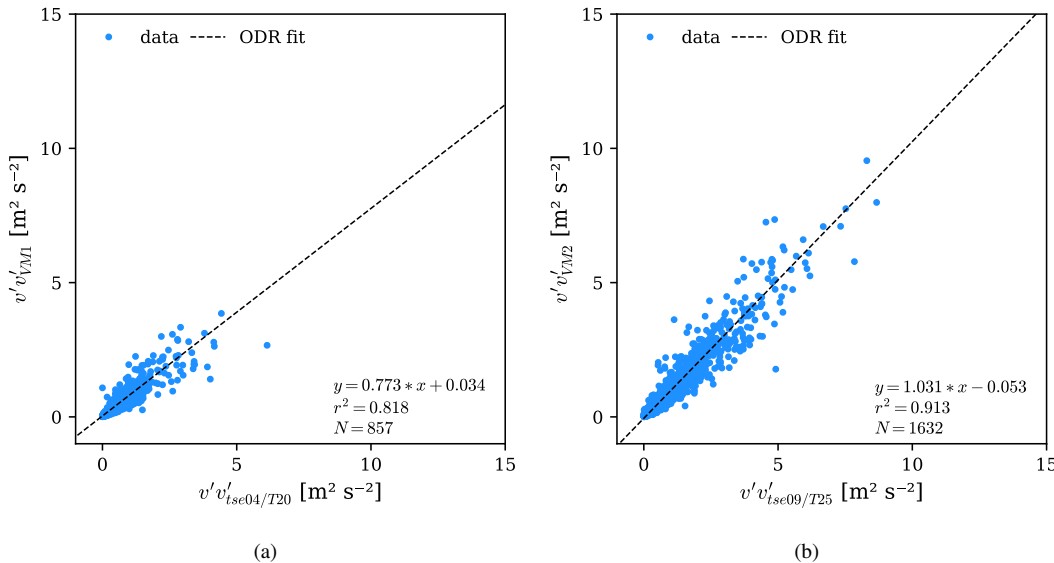

**Figure 8.** Turbulence measurements of virtual masts against sonic anemometer data: (a) VM1 and tse04/T20 $v'v'$ at 80 m a.g.l. and (b) VM2 and tse09/T25 $v'v'$ at 100 m a.g.l.

For $u'u'$ and $v'v'$, the VMs' low sampling rate led to a weaker linear correlation against sonic measurements than for $u$ and $v$. The $r^2$ results, which were higher than 0.987 (VM2 $u$ and $v$) for the mean wind speed components, assumed values as low as 0.809 (VM1 $v'v'$ at 80 m a.g.l.) in the variances (Table 7 and Fig. 8). This means the VM turbulence measurements did not portray the wind variability, represented by $r^2$, as the sonic anemometer readings and the VM averages.

In the linear regression equation between VM and sonic turbulence measurements, $b$ was close to zero in all VMs, with the highest value of $-0.130\,\mathrm{m^2\,s^{-2}}$ for $u'u'$ at VM2; while the slope coefficient ($m$) ranged from 0.799 at 80 m VM1 ($v'v'$) to 1.269 at 100 m VM2 ($u'u'$). The steeper slope for VM2's turbulence measurements (both above 1) indicated greater sensitivity to changes in turbulence compared to the other VMs, where $m$ was less than 1. However, this did not translate into better accuracy, as VM2 had the highest $RMSE$ for turbulence measurements.

Regarding errors, on the ridges, the average $RMSE$ for the turbulent wind components ($0.295\,\mathrm{m^2\,s^{-2}}$) was lower than in the valley ($0.475\,\mathrm{m^2\,s^{-2}}$), as also observed in the radial velocity results. The $RMSE$ at VM2 for turbulence measurements was the highest, $0.600\,\mathrm{m^2\,s^{-2}}$ for $u'u'$; while the highest $Bias$ was at VM3 ($-0.143\,\mathrm{m^2\,s^{-2}}$ for $u'u'$), closely followed by VM2 ($0.131\,\mathrm{m^2\,s^{-2}}$ for $u'u'$), in absolute values. The high errors in VM2 turbulence measurements are attributed to the approximately 9-second mismatch between the lidars. Other contributing factors are the small interception angle between the lidars' beams and the measurement sampling rate, which may be insufficient for the valley complex flow, as also observed in the VM2 mean flow results. Consistently with the distinct valley flow, $u'u'$ measured by VM2 uniquely overestimated the sonic measurements (positive $Bias$), despite the negative $Bias$ in the radial velocity variances of WS2 and WS6.

Overall, the VM turbulence measurements showed a high mean $r^2$ value (0.867) and low mean errors ($0.340\,\mathrm{m^2\,s^{-2}}\;RMSE$

and $-0.063\,\mathrm{m^2\,s^{-2}}\;Bias$), despite the average $r^2$ being lower than that of the mean wind components (0.994), the imperfect synchronisation of the scans, and the low sampling rate. The relatively high accuracy of the VM results in capturing the turbulent flow, even with measurement constraints, indicates that in Perdigão, synoptic and mesoscale systems dominate the atmospheric circulation at the site, and small-scale phenomena played a minor role in the wind patterns.

In Pauscher et al. (2016), the $r^2$ values of $u'u'$ ($v'v'$) were equal to 0.954 (0.966), 0.887 (0.903), and 0.782 (0.861), for the three different dual-lidar combinations. On average, their $r^2$ values were ~1 % (~6 %) higher than the ones depicted here. This difference is again related to the nature of the scans (staring versus RHI combination), which affects the time-spatial synchronisation and the measurement frequency.

## 4.2 Influences on the dual-lidar results

Besides the inherent differences between point-based sonic readings and volumetric-based VM measurements, additional factors can cause the VM results to diverge further from the reference readings. Our analysis focused on three potential factors: atmospheric stability, vertical velocity, and sampling rate.

### 4.2.1 Atmospheric stability

To assess the atmospheric stability influence on mean and turbulence measurements in a multi-lidar setup, we categorised VM1–3 measurements according to the atmospheric stability of the nearby 100 m towers, estimated by the bulk Richardson number ($Ri_B$), similar to Menke et al. (2019b), being assigned as stable ($Ri_B > 0$) or unstable ($Ri_B \leq 0$). While previous studies focused on the stability influence on VMs in flat terrains (Newman et al., 2016; Choukulkar et al., 2017), the virtual masts in Perdigão were located in mountainous terrain, where the complex wind flow can disrupt a direct correlation between stability and dual-lidar measurements.

The bulk Richardson number ($Ri_B$) was calculated with the $10\,\mathrm{min}$ average horizontal mean wind speed components measured at $100\,\mathrm{m}$ a.g.l. ($u_{100}$ and $v_{100}$) and assuming relatively dry air conditions, i.e., using the $10\,\mathrm{min}$ average potential temperature at $2\,\mathrm{m}$ ($\Theta_2$) and $100\,\mathrm{m}$ ($\Theta_{100}$) height rather than the virtual potential temperature (Stull, 1988):

$$Ri_B = \frac{g(\Theta_{100} - \Theta_2)\Delta z}{\Theta_{100}\left[(u_{100})^2 + (v_{100})^2\right]}. \tag{4}$$

The gravitational acceleration is $g = 9.81\,\mathrm{m\,s^{-2}}$, $\Delta z = (100 - 2)$ m, and the wind speed at $2\,\mathrm{m}$ a.g.l. was assumed equal to zero. The $10\,\mathrm{min}$ average potential temperature was approximated by $\Theta \approx T + (g/C_p)z$, where $g/C_p = 0.0098\,\mathrm{K\,m^{-1}}$ and $T$ is the $10\,\mathrm{min}$ average air temperature (Stull, 1988) measured by the temperature sensors.

We assumed relatively dry air conditions ($\Theta_v \approx \Theta$) due to the lack of pressure measurements on Perdigão's $100\,\mathrm{m}$ towers and the limited availability of barometric data from nearby towers, which reduced the number of periods for which we could calculate $Ri_B$ and classify atmospheric stability. This assumption proved valid because the differences between the $10\,\mathrm{min}$ average $\Theta_v$ and the $10\,\mathrm{min}$ average $\Theta$ at the three $100\,\mathrm{m}$ towers did not exceed $3.8\,\mathrm{K}$ at $100\,\mathrm{m}$ a.g.l. during the entire IOP.

The distribution of the $Ri_B$ values at the three $100\,\mathrm{m}$ towers (Fig. 9) further highlights the different conditions between ridge and valley wind flow. For tse04/T20 and tse13/T29, the histograms peak around zero $Ri_B$ with nearly symmetrical distributions, showing similar quantities of unstable and stable conditions. The valley tower, on the other hand, has a broader distribution with a significant spread towards positive $Ri_B$ values, indicating greater variability in stability compared to the ridge towers and a prevalence of stable atmospheric conditions.

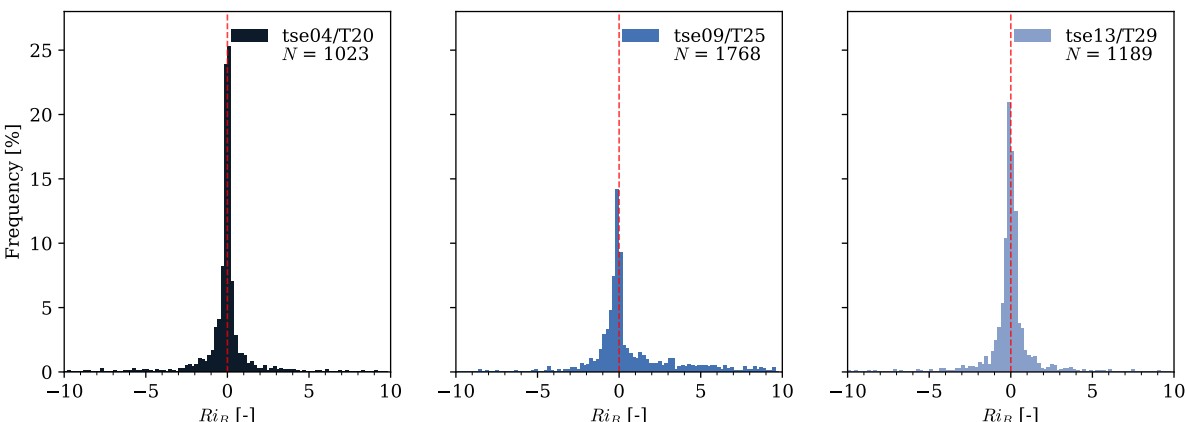

**Figure 9.** Histogram of the bulk Richardson number from $10\,\mathrm{min}$ average measurements at tse04/T20 (SW ridge), tse09/T25 (valley), and tse13/T29 (NE ridge) at $100\,\mathrm{m}$ a.g.l. during the VM measurement periods of the mean flow. The $Ri_B$ values are constrained to the -10 to 10 interval, with a bin width of 0.2.

From the data collected by the $100\,\mathrm{m}$ towers, the following number of $10\,\mathrm{min}$ periods were classified as unstable (stable) at VM1–3: 526 (497), 780 (988), and 617 (572) for the mean wind components at $100\,\mathrm{m}$ a.g.l. For the variances, the respective quantities were 447 (383) at VM1, 719 (898) at VM2, and 552 (514) at VM3.

The influence of atmospheric stability on the dual-lidar results was affected by the distinct wind flows between the ridges and the valley in Perdigão (Table 8), as well as by the different spatial (WSs' interception angle) and temporal (WSs' desyn-
chronisation) configurations among the VMs. On the ridges, VM1 and VM3 showed slightly better $r^2$ values and slightly lower errors under stable than unstable atmospheric conditions, especially for turbulent flow variables. The average $r^2$, $RMSE$, and $Bias$ for the mean wind components ($u$ and $v$) were 0.997, $0.414\,\mathrm{m\,s^{-1}}$, and $0.082\,\mathrm{m\,s^{-1}}$ in stable conditions; while under unstable conditions, these were equal to 0.996, $0.434\,\mathrm{m\,s^{-1}}$ and $0.075\,\mathrm{m\,s^{-1}}$. For turbulence variables ($u'u'$ and $v'v'$), the statistical metrics assumed mean values of 0.853, $0.235\,\mathrm{m^2\,s^{-2}}$, and $-0.055\,\mathrm{m^2\,s^{-2}}$ for stable, and 0.836, $0.339\,\mathrm{m^2\,s^{-2}}$, and
$-0.140\,\mathrm{m^2\,s^{-2}}$ for unstable conditions.

Conversely, at the valley VM, higher $r^2$ values and lower errors with a stable atmosphere were restricted to $u'u'$ and $v'v'$. The variances $r^2$, $RMSE$, and $Bias$ under stable conditions were 0.891, $0.358\,\mathrm{m^2\,s^{-2}}$, and $0.029\,\mathrm{m^2\,s^{-2}}$, on average. In comparison, the average $u'u'$ and $v'v'$ metrics during unstable conditions were equal to 0.827, $0.587\,\mathrm{m^2\,s^{-2}}$, and $0.085\,\mathrm{m^2\,s^{-2}}$.

**Table 8.** Statistical parameters from VM and tower comparisons according to the atmospheric stability.

| | Height a.g.l. [m] | Metric | Stability | Variables | | | | |
| --- | --- | --- | --- | --- | --- | --- | --- | --- |
| | | | | $u$ | $v$ | $V_h$ | $u'u'$ | $v'v'$ |
| VM1 | 80 | $r^2$ | unstable | 0.998 | 0.993 | 0.977 | 0.879 | 0.775 |
| | | | stable | 0.998 | 0.997 | 0.985 | 0.784 | 0.801 |
| | | $RMSE$ | unstable | 0.405 | 0.395 | 0.368 | 0.259 | 0.368 |
| | | | stable | 0.390 | 0.358 | 0.313 | 0.147 | 0.253 |
| | | $Bias$ | unstable | 0.357 | −0.292 | 0.101 | −0.132 | −0.139 |
| | | | stable | 0.354 | −0.287 | 0.120 | −0.047 | −0.063 |
| | 100 | $r^2$ | unstable | 0.998 | 0.994 | 0.976 | 0.863 | 0.771 |
| | | | stable | 0.998 | 0.996 | 0.987 | 0.826 | 0.845 |
| | | $RMSE$ | unstable | 0.437 | 0.410 | 0.392 | 0.257 | 0.352 |
| | | | stable | 0.398 | 0.398 | 0.319 | 0.141 | 0.302 |
| | | $Bias$ | unstable | 0.389 | −0.317 | 0.106 | −0.121 | −0.133 |
| | | | stable | 0.364 | −0.310 | 0.099 | −0.035 | −0.063 |
| VM2 | 100 | $r^2$ | unstable | 0.990 | 0.987 | 0.954 | 0.771 | 0.882 |
| | | | stable | 0.983 | 0.987 | 0.940 | 0.854 | 0.928 |
| | | $RMSE$ | unstable | 0.516 | 0.269 | 0.430 | 0.729 | 0.444 |
| | | | stable | 0.561 | 0.323 | 0.454 | 0.471 | 0.245 |
| | | $Bias$ | unstable | 0.455 | −0.097 | 0.141 | 0.204 | −0.034 |
| | | | stable | 0.517 | −0.202 | 0.225 | 0.072 | −0.013 |
| VM3 | 100 | $r^2$ | unstable | 0.995 | 0.996 | 0.959 | 0.863 | 0.869 |
| | | | stable | 0.995 | 0.998 | 0.971 | 0.940 | 0.924 |
| | | $RMSE$ | unstable | 0.611 | 0.344 | 0.506 | 0.416 | 0.381 |
| | | | stable | 0.649 | 0.291 | 0.416 | 0.294 | 0.272 |
| | | $Bias$ | unstable | 0.555 | −0.245 | 0.147 | −0.208 | −0.109 |
| | | | stable | 0.603 | −0.231 | 0.148 | −0.075 | −0.048 |

The $RMSE$ and $Bias$ units are [$\mathrm{m\,s^{-1}}$] for $u$, $v$, $V_h$ variables, while for $u'u'$ and $v'v'$ are [$\mathrm{m^2\,s^{-2}}$]. $r^2$ is unitless.

Another distinct result at VM2 was that regardless of the atmospheric conditions, the $u'u'$ turbulence measurement overestimated the tse09/T25 sonic anemometer readings at $100\,\mathrm{m}$ a.g.l.

The overall better results from the VMs under stable than unstable atmospheric conditions indicate that when the air is more stable and less turbulent, the temporal and spatial synchronisation between the scans of a multi-lidar system becomes less critical, without compromising the accuracy of the measurements. Additionally, while the statistical metrics for the $10\,\mathrm{min}$ mean values changed slightly according to stability, the metrics for the $10\,\mathrm{min}$ variances were more affected by atmospheric

conditions. In terms of wind direction, there was no clear relationship between the VM wind direction error (i.e., the difference between the VM's 10 min average horizontal wind direction and the tower's 10 min average horizontal wind direction) and atmospheric stability (not shown here).

### 4.2.2 Vertical velocity

Another possible influence on VM retrievals was the assumption of a zero vertical wind velocity ($w$) made to obtain the horizontal wind components from the WindScanners' radial velocities (Step 4 in Sec. 3.1). The coefficient of determination of the linear regression between the 10 min average $w$ values measured by sonic anemometers and the 10 min horizontal wind speed errors of the VMs (i.e., the difference between the VM's 10 min average horizontal wind speed and the anemometer's 10 min average horizontal wind speed) around 100 m a.g.l. in Perdigão was lower than 0.060 at all measurement locations. For turbulence measurements, the highest $r^2$ between the 10 min average $w$ values and the 10 min VM measurement errors was 0.110 at VM1.

These low $r^2$ values mean that the assumption of zero vertical wind velocity had a minimal impact on the VM measurements at 80 and 100 m a.g.l. in Perdigão, confirming the validity of the VM results at these heights. This minimal impact is attributed to the small elevation angles of the lidars' beams (Table 2) and the low vertical velocity at the site, which did not exceed $3.6\,\mathrm{m\,s}^{-1}$ at 100 m a.g.l. during the IOP.

At heights above 100 m, however, the elevation angles of the beams will be higher, causing the lidar beams to be more aligned with the vertical component of the wind. Thus, in a strong convective atmosphere at higher heights, the vertical velocity can influence the virtual-mast results more significantly. In Perdigão, the maximum elevation angles of the VMs were: 21.6° at VM1, 15.6° at VM2, and 23.0° at VM3.

### 4.2.3 Sampling rate

We turned to the sonic data to assess how the VM sampling rate affected the results. Results at progressively lower sampling rates were compared against the 20 Hz measurements in terms of $r^2$, $RMSE$ (Fig. 10), and $Bias$. The data were down-sampled by selecting every $n$-th sample for frequencies between 1 Hz and 20 Hz (e.g., for 2 Hz, every 10th sample), and by selecting the $n$-th time step for frequencies below 1 Hz (e.g., for 0.5 Hz, every 2nd time step). Following down-sampling, variances and averages were calculated over 10 min intervals. Then, to assess the influence of the sampling rate in the VM retrievals, the statistical metrics of the sonic data were linearly interpolated at the VMs' acquisition rates, between 0.018–0.038 Hz for the means and 0.030–0.038 Hz for the variances (shaded area in Fig. 10).

Similar to the previous results, the mean wind flow ($u$, $v$, and $V_h$) and the metrics $r^2$ and $Bias$ showed less sensitivity to measurement frequency than the variances ($u'u'$ and $v'v'$) and $RMSE$ at the three 100 m towers. Additionally, the sampling rate had a similar influence on the wind components of the same moment, evidenced by the comparable results for $u$ and $v$ and for $u'u'$ and $v'v'$ at Table 9. Consequently, Figure 10 displays only the $RMSE$ for mean and turbulent $x$-axis wind speed component at 100 m a.g.l.

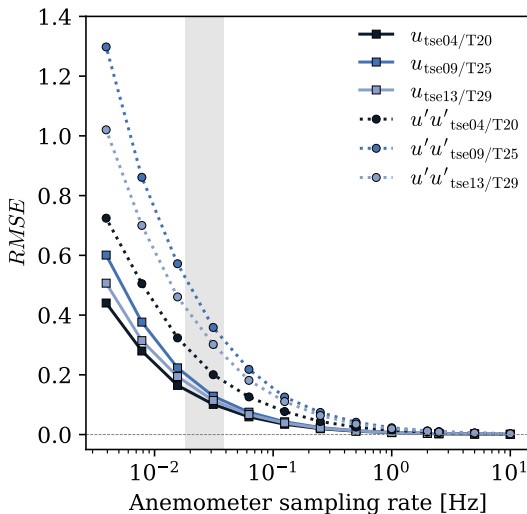

**Figure 10.** $RMSE$ of sonic measurements by the sampling rate, for the mean ($u$) and turbulent ($u'u'$) $x$-axis wind speed component, on the three $100\,\mathrm{m}$ towers at $100\,\mathrm{m}$ a.g.l. The $RMSE$ units are $[\mathrm{m\,s^{-1}}]$ for $u$, and $[\mathrm{m^2\,s^{-2}}]$ for $u'u'$.

**Table 9.** Averaged statistical metrics due to sampling rates in the virtual-mast measurement range for the mean (0.018–0.038 Hz) and turbulent (0.030–0.038 Hz) flow, based on sonic readings at $100\,\mathrm{m}$ a.g.l.

| Metric | Mean flow | | | Turbulent flow | |
|---|---|---|---|---|---|
| | $u$ | $v$ | $V_h$ | $u'u'$ | $v'v'$ |
| $r^2$ | 0.995–0.998 | 0.996–0.999 | 0.992–0.997 | 0.911–0.931 | 0.930–0.945 |
| $RMSE$ | 0.104–0.180 | 0.104–0.179 | 0.102–0.178 | 0.262 – 0.300 | 0.267–0.306 |
| $Bias$ | 0.001–0.002 | $\sim$0–$-$0.001 | 0.003–0.008 | $-$0.012–$-$0.017 | $-$0.011–$-$0.015 |

The units of $RMSE$ and $Bias$ are $[\mathrm{m\,s^{-1}}]$ for mean variables, while for variances are $[\mathrm{m^2\,s^{-2}}]$. $r^2$ is unitless.

At $100\,\mathrm{m}$ a.g.l., the estimated average $RMSE$ of the VMs, due solely to their sampling rate, ranged between 0.102 and 0.180 $\mathrm{m\,s^{-1}}$ for the mean flow quantities and 0.262 and 0.306 $\mathrm{m^2\,s^{-2}}$ for the turbulence variables (Table 9). Considering the overall $RMSE$ values for all virtual masts at $100\,\mathrm{m}$ a.g.l. (0.434 $\mathrm{m\,s^{-1}}$ for the average of $u$ and $v$ and 0.363 $\mathrm{m^2\,s^{-2}}$ for the average of $u'u'$ and $v'v'$), around 33 % of the VMs' $RMSE$ for the mean wind components and 78 % for the variances can be attributed to their measurement frequency, assuming a linear influence of this factor. For the mean horizontal wind velocity, 33 % of the VMs' average $RMSE$ at $100\,\mathrm{m}$ a.g.l. can be attributed to their measurement frequency. Additionally, to accurately measure the wind flow in the valley, a higher sampling rate is required than above the hills, especially to retrieve the wind variances. Within the VM sampling rate range, the average $RMSE$ error for turbulence measurements at $100\,\mathrm{m}$ a.g.l. is about 61 % and 19 % higher in the valley than on the SW and NE ridge.

Therefore, when aiming for dual-lidar readings with errors due to the sampling rate lower than those presented here, one should evaluate the elevation range covered in the RHI mode, the lidar's acquisition time, and the type of scan. Additionally, the influence of the sampling rate on measurements should be considered when planning new experimental campaigns, particularly in the selection of equipment and measurement frequency of targeted wind variables. For a mininum $RMSE$ increase (below $0.1 \, \text{m} \, \text{s}^{-1}$ and $0.1 \, \text{m}^2 \, \text{s}^{-2}$) compared to the $20 \, \text{Hz}$ frequency, the VM sampling rate should be at least $0.05 \, \text{Hz}$ for mean quantities and $0.2 \, \text{Hz}$ for turbulence measurements.

## 5  Conclusions

Dual-lidar measurements of Range Height Indicator (RHI) scans in a virtual mast (VM) mode were compared against sonic anemometer readings at three $100 \, \text{m}$ towers over the Perdigão complex terrain, to evaluate the VM measurement uncertainty and validate its use over large distances above the ground. The study focused on $10 \, \text{min}$ means and variances of radial velocity ($v_r$), wind speed ($V_h$), and wind velocity ($u$ and $v$), retrieved by dual-lidar and sonic anemometers at $80 \, \text{m}$ and $100 \, \text{m}$ a.g.l. A methodology for processing the virtual mast dataset was also devised.

In the analysis of the mean flow, a high correlation was found between VM and sonic measurements, with $r^2$ values close to 1 at all VMs. Notably, the lowest $r^2$ were observed at VM2 ($0.987$ for $u$ and $v$, and $0.948$ for $V_h$), attributed to the small angle ($\sim 40.2°$) between the lidars' beams (leading to high dual-lidar error propagation) and to the more turbulent flow in the valley. Regarding the errors, the average $RMSE$ and $Bias$ for $u$ and $v$ was $0.422 \, \text{m} \, \text{s}^{-1}$ and $0.102 \, \text{m} \, \text{s}^{-1}$ for all VMs, with the highest values occurring at VM3, $0.626 \, \text{m} \, \text{s}^{-1}$ and $0.575 \, \text{m} \, \text{s}^{-1}$, for the $u$ component. The error magnitudes were consistent for all mean flow variables ($u$, $v$, and $V_h$) within each virtual mast. However, the average $r^2$ for $V_h$ ($0.969$) was lower than for the wind components ($0.994$).

The low measuring frequency ($0.038 \, \text{Hz}$ maximum) and the VM location mainly impacted the turbulence measurements ($u'u'$ and $v'v'$). The average $r^2$ that was equal to $0.994$ for the mean wind components, was $0.867$ for the variances. In the linear regression equation, the constants ($b$) took on values close to zero for all VMs, while the slope coefficients ($m$) varied from $0.799$ for $v'v'$ VM1 to $1.269$ for $u'u'$ VM2. The greater sensitivity of VM2 to turbulence changes, however, did not translate into better accuracy. The $RMSE$ for $u'u'$ and $v'v'$ across all VMs averaged $0.340 \, \text{m}^2 \, \text{s}^{-2}$, with the highest value observed in the valley (VM2), reaching $0.600 \, \text{m}^2 \, \text{s}^{-2}$ for $u'u'$, due to the worse lidars' synchronisation (about $9 \, \text{s}$), the smaller between-beam angle, and the complex valley flow. Overall, the VM correlations against reference turbulence measurements were still high and the average errors were low ($0.340 \, \text{m}^2 \, \text{s}^{-2}$ $RMSE$ and $-0.063 \, \text{m}^2 \, \text{s}^{-2}$ $Bias$), indicating that small-scale phenomena play a smaller role at $80 \, \text{m}$ and $100 \, \text{m}$ a.g.l. in Perdigão.

The influence of atmospheric stability also depended on the VM location. The virtual masts on the ridges (VM1 and VM3) showed higher correlations and lower errors under stable than unstable conditions. Namely for the variances, where the average $r^2$, $RMSE$, and $Bias$ for VM1 and VM3 under stable (unstable) conditions were equal to $0.853$ ($0.836$), $0.235 \, \text{m}^2 \, \text{s}^{-2}$ ($0.339 \, \text{m}^2 \, \text{s}^{-2}$), and $-0.055 \, \text{m}^2 \, \text{s}^{-2}$ ($-0.140 \, \text{m}^2 \, \text{s}^{-2}$). In the valley (VM2), the better statistical metrics with stable conditions were restricted to the variance measurements of the wind; showing average $r^2$, $RMSE$, and $Bias$ of $0.891$ ($0.827$),

$0.358\,\mathrm{m^2\,s^{-2}}$ ($0.587\,\mathrm{m^2\,s^{-2}}$), and $0.029\,\mathrm{m^2\,s^{-2}}$ ($0.085\,\mathrm{m^2\,s^{-2}}$) with stable (unstable) atmosphere. Although atmospheric stability differently affected the accuracy of VM measurements on the ridges and in the valley, the results indicate that in a stable, less turbulent atmosphere, synchronisation between the scans of a multi-lidar system becomes less critical for maintaining measurement accuracy than in unstable conditions. Regarding the VM wind direction, no correlation between its errors and atmospheric stability could be drawn.

The impact of the zero vertical velocity assumption on dual-lidar retrievals at $80$ and $100\,\mathrm{m}$ a.g.l. in Perdigão was minimal, confirming the validity of the VM results at these heights. The $r^2$ results were lower than $0.060$ for the $10\,\mathrm{min}$ average $w$ values from sonic anemometer readings and the $10\,\mathrm{min}$ horizontal wind speed errors from the VM measurements, and lower than $0.110$ for the $10\,\mathrm{min}$ average $w$ values and the $10\,\mathrm{min}$ variance errors from the VMs.

Lastly, the influence of the VM sampling rate accounted for $33\,\%$ of the overall $RMSE$ for the mean quantities and $78\,\%$ for the variances at $100\,\mathrm{m}$ a.g.l. when assuming a linear influence of this factor on the dual-lidar error. The impact of sampling rate on measurements, including those from dual-lidars, is crucial when selecting and configuring equipment to ensure accurate recording of target variables.

Overall, Perdigão's VMs obtained accurate mean flow measurements, and their turbulence estimations, despite displaying lower correlations against reference data, also showed low errors, demonstrating the VMs' ability to capture mean and turbulent wind characteristics under different flow conditions, at great heights, and in complex terrain. From the VM measurements and sonic readings, the construction of vertical profiles of the wind enables the analysis of Perdigão's complex flow at heights up to $430\,\mathrm{m}$ a.g.l.

For greater data accuracy and reliability in future dual-lidar campaigns, the lidars must be positioned to form an approximately $90°$ angle between their beams to minimise error propagation and operated at a sampling frequency of at least $0.05\,\mathrm{Hz}$ for mean quantities and $0.2\,\mathrm{Hz}$ for turbulence. These frequencies yield a minimal $RMSE$ increase (below $0.1\,\mathrm{m\,s^{-1}}$ and $0.1\,\mathrm{m^2\,s^{-2}}$) compared to the $20\,\mathrm{Hz}$ frequency.

*Data availability.* The data collected during the Perdigão-2017 campaign are available in the websites: https://perdigao.fe.up.pt/ (University of Porto, 2020) and https://data.eol.ucar.edu/master_lists/generated/perdigao/ (UCAR/NCAR - Earth Observing Laboratory, 2019a). The lidar data measured by the Technical University of Denmark is also available at: https://doi.org/10.11583/DTU.7228544.v1 (Menke et al., 2018).

*Author contributions.* Isadora Coimbra and Vasco Batista developed the initial methodology, validation, and code. Isadora Coimbra refined the methodology, further processed and analysed the data, and wrote the first manuscript draft under the guidance of Jakob Mann and José Palma. All authors participated actively in reviewing the work and rewriting multiple versions of the manuscript.

*Competing interests.* The authors declare that they have no conflict of interest.

*Acknowledgements.* This research was carried out within the scope of the LIdar Knowledge Europe (LIKE) project, H2020-MSCA-ITN-2019, funded by the European Union, Grant no. 858358. We thank the anonymous referee and Joachim Reuder for their valuable feedback and suggestions.

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
