# Peer review of "Exploring dual-lidar mean and turbulence measurements over Perdigão's complex terrain"

_EGUsphere, 2024_

## Referee Comment (RC1)

Exploring dual-lidar mean and turbulence measurements over complex terrain

Isadora L. Coimbra, Jakob Mann, and José M.L.M. Palma

**General comments**

This manuscript investigates a specific type of wind and turbulence measurement by two Doppler lidars, namely by forming a virtual mast by overlapping two coordinated Range Height Indicator (RHI) scans. Aim is provide vertical profiles of the wind and turbulence by remote sensing, i.e. without the need of a tall tower and might therefore be more cost efficient, more flexible and able to cover higher altitude.

To test this method, these dual-lidar measurements are compared with in-situ mast measurements (sonic anemometers) in a very complex environment, in general not suitable for single lidar measurements (in particular regarding turbulence), as homogenous flow conditions cannot be assumed. For this study, measurement data from a well-known Perdigao-2017 campaign is used.

Overall, the manuscript is very well written and structured. The introduction covers the many layers in terms multiple Doppler lidars usages, type of scans, type of terrain, and type of intercomparisons. As such it is clear where to put this study. The campaign and instruments are well introduced and constraints and error sources of the dual-lidar measurements are well explained, providing the relevant formulae. The results are well presented, both in graphs and in tables. This manuscripts provides a real, quantitative picture on how well two coordinated Doppler lidars can provide wind and turbulence in a real complex terrain. Also, the recommendations of the minimal sampling rate are very valuable.

I do have some minor and slightly larger comments. My main comments are:

(A) Abstract", page 1, line 21: "Upon appraisal of the VM accuracy based on sonic anemometer measurements at 80 and 100 m a.g.l., we obtained vertical wind profiles up to 430 m a.g.l."

This point does not really come back in the remaining of the manuscript. Would it be possible to show some examples or interesting cases, in which the ability to measure beyond the mast size becomes very clear?

(B) Page 7, line 156: "Upon validating their accuracy, we can use the entire dataset in further studies, assuming that the accuracy is consistent at higher levels."

The assumption of zero vertical velocity becomes more stringent for larger elevation angles (higher levels), as the vertical component of the measured radial velocity becomes larger. As such, I am not sure whether the extrapolation conclusions made a basis of a given altitude to higher altitude can simply be done. I am not convinced that one can assume that the accuracy at 80m or 100m will be the same at 400m. I think the role of elevation angle, and the increasing vertical component of measured radial velocity (or the deceasing cos(phi) terms in Eq. (1)) should at least be mentioned in this discussion.

(C) page 20, paragraph 4.2.2. Vertical velocity

In general I think this part is way too short. Especially the sentence "However, no correlation was observed between the w values measured by sonic anemometers and the horizontal wind speed errors of the VMs" brings up many questions. First, what "w values" do you mean? 10-minute averages, nearest sample value, 10-minute variances? Considering the very local behavior of up- and downdraft and turbulence one has to be very careful in this comparison, e.g. considering the spatial

mismatch between VMs and sonics. Conclusions based on a simple correlation might not be sufficient. And how do you quantify "no correlation"? Second, why only considering horizontal wind speed? Why would that be representative for the other variables (or why it would be the most sensitive)?

The assumption of zero vertical velocity is the only assumption in Eq. (1), and a major assumption in a dual-lidar virtual mast approach. I agree that with small elevation angles this assumption can be justified, although still in convective conditions with strong updrafts in combination with low wind speeds the vertical component of the radial velocity can be significant. I think it is important to stress that the conclusions drawn in this section are based on those elevation angles corresponding to virtual mast levels of 80m or 100m, but whether they are still true for 400m remains to be seen.

**Specific comments**

(a) Abstract page 1, line 22 and page 22, line 428: "vertical wind profiles"

I find this way of phrasing very confusing. Does it mean profiles of vertical wind or vertical profiles of wind? I guess you mean the second one, but please use a less ambiguous way of describing what you mean.

(b) Page 5, line 113: "Thermohygrometer sensors were installed at seven levels: 2 m, 10 m, 20 m, 40 m, 60 m, 80 m, and 100 m a.g.l".

Thermohygrometer might not be a very commonly known term. Maybe explicitly mentioning "temperature" and "relative humidity" sensors would be better. Also, at this point it is not motivated why these measurements are important for this study. Maybe already introduce their role in this study. Finally, you might want to provide more details on this instrument (manufacturer, type), on the same footing as the sonic anemometer.

(c) Page 15, Table 7: Repeat the meaning of the symbols m and b, for instance by providing again the fit formula (as provided in the main text). Also, one could add at the bottom "m is unitless".

(d) Page 18, line 332: In the definition of the Richardson number (gradient or bulk), as given by Stull 1988 (section 5.6.2 and 5.6.3) that is also used as a reference here, the virtual potential temperature is used, not the potential temperature. This needs to be corrected. By the way, the "thermohygrometer" provides all the means to derive the virtual potential temperature.

(e) Page 18, line 332: "converting the mean temperature into potential temperature". Why "mean" is used in this sentence (or not twice: mean temperature to mean potential temperature)? The time averaging of the temperature data, and the conversion to (virtual) potential temperature are two separate steps. Only in the next paragraph it becomes clear that with mean temperature probably 10-minutes averaged temperature is meant.

**Technical corrections**

(f) Page 11, line 213: "…except for the y-wind component measured by VM1." I guess "y-wind component" is a typo here, because throughout the manuscript u- and v-components are used.

(g) Color usage in the various figures. The different types of blue is hard to distinguish, which is an issue for Figures 5 and 9.

---

## Author Response (AR1)

**Exploring dual-lidar mean and turbulence measurements over Perdigão's complex terrain**

authored by

Isadora L. Coimbra, Jakob Mann, José M. L. M. Palma, and Vasco T. P. Batista.

Submitted for publication in the *Atmospheric Measurement Techniques (AMT)*
EGUsphere [preprint], `https://doi.org/10.5194/egusphere-2024-936`, 2024

**Contents**

**Responses to comments by Reviewer 1 (Anonymous)**

**NOTE:** Our responses to the Reviewer's comments appear in the 'RESPONSE' grey-shaded text box, and the changes in the manuscript are shown in the 'CHANGES' blue-shaded text box.

**General comments**

This manuscript investigates a specific type of wind and turbulence measurement by two Doppler lidars, namely by forming a virtual mast by overlapping two coordinated Range Height Indicator (RHI) scans. Aim is provide vertical profiles of the wind and turbulence by remote sensing, i.e. without the need of a tall tower and might therefore be more cost efficient, more flexible and able to cover higher altitude.

To test this method, these dual-lidar measurements are compared with in-situ mast measurements (sonic anemometers) in a very complex environment, in general not suitable for single lidar measurements (in particular regarding turbulence), as homogenous flow conditions cannot be assumed. For this study, measurement data from a well-known Perdigao-2017 campaign is used.

Overall, the manuscript is very well written and structured. The introduction covers the many layers in terms multiple Doppler lidars usages, type of scans, type of terrain, and type of inter-comparisons. As such it is clear where to put this study. The campaign and instruments are well introduced and constraints and error sources of the dual-lidar measurements are well explained, providing the relevant formulae. The results are well presented, both in graphs and in tables. This manuscripts provides a real, quantitative picture on how well two coordinated Doppler lidars can provide wind and turbulence in a real complex terrain. Also, the recommendations of the minimal sampling rate are very valuable.

I do have some minor and slightly larger comments.

My main comments are:

(A) Abstract", page 1, line 21: "Upon appraisal of the VM accuracy based on sonic anemometer measurements at 80 and 100 m a.g.l., we obtained vertical wind profiles up to 430 m a.g.l."

This point does not really come back in the remaining of the manuscript. Would it be possible to show some examples or interesting cases, in which the ability to measure beyond the mast size becomes very clear?

RESPONSE:

The primary objective of this paper is to explore and understand the capabilities and limitations of the virtual mast (VM) measurements by comparing them against anemometric measurements. While we mentioned the potential to obtain vertical profiles of the wind up to 430 m a.g.l., this study focused on validating the VM measurements at 80 and 100 m a.g.l., as these were the heights for which we had corresponding anemometric measurements.

(B) Page 7, line 156: "Upon validating their accuracy, we can use the entire dataset in further studies, assuming that the accuracy is consistent at higher levels."

The assumption of zero vertical velocity becomes more stringent for larger elevation angles (higher levels), as the vertical component of the measured radial velocity becomes larger. As such, I am not sure whether the extrapolation conclusions made a basis of a given altitude to higher altitude can simply be done. I am not convinced that one can assume that the accuracy at 80m or 100m will be the same at 400m. I think the role of elevation angle, and the increasing vertical component of measured radial velocity (or the deceasing cos(phi) terms in Eq. (1)) should at least be mentioned in this discussion.

RESPONSE:

We acknowledge the concerns regarding the assumption of zero vertical velocity being less valid at higher levels, which can lead to higher errors in retrieving the $u-$ and $v-$ wind components. However, we had no anemometer measurements above 100 m, therefore, we could not evaluate the vertical velocity influence on the VM results above this height.

We have included in the text the potential for increased errors at heights beyond 100 m due to higher beam elevation angles (line 163).

CHANGES:

line 163: "Upon validating their accuracy, we can use the entire VM dataset in further studies. However, at higher heights, the assumption of zero vertical velocity (Step 4) can reduce the accuracy of the horizontal wind components obtained from dual-lidar measurements, since the increase in beam elevation angles causes the lidar beams to be more aligned with the vertical component of the wind."

(C) page 20, paragraph 4.2.2. Vertical velocity

In general I think this part is way too short. Especially the sentence "However, no correlation was observed between the w values measured by sonic anemometers and the horizontal wind speed

errors of the VMs" brings up many questions. First, what "w values" do you mean? 10-minute averages, nearest sample value, 10-minute variances? Considering the very local behavior of up- and downdraft and turbulence one has to be very careful in this comparison, e.g. considering the spatial mismatch between VMs and sonics. Conclusions based on a simple correlation might not be sufficient. And how do you quantify "no correlation"? Second, why only considering horizontal wind speed? Why would that be representative for the other variables (or why it would be the most sensitive)?

The assumption of zero vertical velocity is the only assumption in Eq. (1), and a major assumption in a dual-lidar virtual mast approach. I agree that with small elevation angles this assumption can be justified, although still in convective conditions with strong updrafts in combination with low wind speeds the vertical component of the radial velocity can be significant. I think it is important to stress that the conclusions drawn in this section are based on those elevation angles corresponding to virtual mast levels of 80m or 100m, but whether they are still true for 400m remains to be seen.
* * *
RESPONSE:

The mentioned "w values" are 10 min averages of the vertical velocity component measured by the sonic anemometers. This information has been included in the revised manuscript (line 385).

At first, we focused on horizontal wind speed in the text due to its importance in wind energy applications. We have now included the $r^2$ values for turbulence measurements as well (line 388). However, regardless of the flow measurement, the $r^2$ between VM's 10 min measurement error and the 10 min average vertical velocity did not exceed 0.110.

Lastly, we have emphasised that our conclusions are based on the VM measurements at 80 and 100 m a.g.l. (paragraph starting at line 391) and that at higher heights, the vertical velocity can influence the measurements more significantly (paragraph starting at line 395).
* * *
CHANGES:

line 385: "The coefficient of determination of the linear regression between the 10 min average $w$ values measured by sonic anemometers and the 10 min horizontal wind speed errors of the VMs (i.e., the difference between the VM's 10 min average horizontal wind speed and the anemometer's 10 min average horizontal wind speed) around 100 m a.g.l. in Perdigão was lower than 0.060 at all measurement locations."

line 388: "For turbulence measurements, the highest $r$ between 10 min VM measurement errors and the 10 min average $w$ values was 0.110 at VM1."

line 391: "These low $r$ values mean that the assumption of zero vertical wind

velocity had a minimal impact on the VM measurements at 80 and 100 m a.g.l. in Perdigão, confirming the validity of the VM results at these heights."

line 395: "At heights above 100 m, however, the elevation angles of the beams will be higher, causing the lidar beams to be more aligned with the vertical component of the wind. Thus, in a strong convective atmosphere at higher heights, the vertical velocity can influence the virtual-mast results more significantly."

**Specific comments**

(a) Abstract page 1, line 22 and page 22, line 428: "vertical wind profiles"

I find this way of phrasing very confusing. Does it mean profiles of vertical wind or vertical profiles of wind? I guess you mean the second one, but please use a less ambiguous way of describing what you mean.

> RESPONSE:
>
> We intended to convey "vertical profiles of the wind". We have replaced "vertical wind profiles" with "vertical profiles of the wind" in the manuscript (lines 22 and 470).

(b) Page 5, line 113: "Thermohygrometer sensors were installed at seven levels: 2 m, 10 m, 20 m, 40 m, 60 m, 80 m, and 100 m a.g.l".

Thermohygrometer might not be a very commonly known term. Maybe explicitly mentioning "temperature" and "relative humidity" sensors would be better. Also, at this point it is not motivated why these measurements are important for this study. Maybe already introduce their role in this study. Finally, you might want to provide more details on this instrument (manufacturer, type), on the same footing as the sonic anemometer.

> RESPONSE:
>
> We have replaced "thermohygrometer" with "temperature/humidity sensor" (line 116). Additionally, we introduced the importance of these measurements earlier in Section 2.2 (line 112) and included more details about the temperature/humidity sensor (line 116).

CHANGES:

line 116: "NCAR SHT75 temperature/humidity sensors were installed at seven levels: $2\,\text{m}$, $10\,\text{m}$, $20\,\text{m}$, $40\,\text{m}$, $60\,\text{m}$, $80\,\text{m}$, and $100\,\text{m}$ a.g.l."

line 112: "The tower equipment provided wind speed and temperature measurements that were used in this study to evaluate the VM wind speed retrievals and classify the atmospheric stability. "

(c) Page 15, Table 7: Repeat the meaning of the symbols m and b, for instance by providing again the fit formula (as provided in the main text). Also, one could add at the bottom "m is unitless".

RESPONSE:

We revised Table 7 (page 15) to include the meanings of the symbols $m$ and $b$, and that $m$ is unitless in the footnote.

CHANGES:

Table 7 footnote: "$m$ and $r^2$ are unitless. $m$ is the coefficient, and $b$ is the constant of the linear regression equation $(y = mx + b)$."

(d) Page 18, line 332: In the definition of the Richardson number (gradient or bulk), as given by Stull 1988 (section 5.6.2 and 5.6.3) that is also used as a reference here, the virtual potential temperature is used, not the potential temperature. This needs to be corrected. By the way, the "thermohygrometer" provides all the means to derive the virtual potential temperature.

RESPONSE:

The equation is indeed for the bulk Richardson number, which we have corrected in the manuscript (lines 338 and 343).

In response to the reviewer's suggestion, we have recalculated the bulk Richardson number $(Ri_B)$ using the virtual potential temperature $(\Theta_v)$ at $2\,\text{m}$ $(\Theta_{v_2})$ and $100\,\text{m}$ $(\Theta_{v_{100}})$ height, and the horizontal mean wind components measured at $100\,\text{m}$ a.g.l. $(u_{100}$ and $v_{100})$ (Stull, 1988):

$$Ri_B = \frac{g(\Theta_{v_{100}} - \Theta_{v_2})\Delta z}{\Theta_{v_{100}} \left[(u_{100})^2 + (v_{100})^2\right]}. \tag{1}$$

The gravitational acceleration is $g = 9.81\,\text{m}\,\text{s}^{-2}$, $\Delta z = (100 - 2)$ m, and the wind speed at $2\,\text{m}$ a.g.l. was assumed equal to zero. All values obtained from the measured fields (velocity, temperature, relative humidity, pressure) were 10-min averaged before calculating the derived quantities. For simplicity, we

forego the representation of the time-averaging operator (e.g., for potential temperature $\Theta \equiv \overline{\Theta}$). The virtual potential temperature was determined using the relation

$$\Theta_v = \Theta(1 + 0.61r), \tag{2}$$

where $\Theta$ is the potential temperature and $r$ is the water-vapor mixing ratio of the air.

The potential temperature was calculated by:

$$\Theta = T \left( \frac{P_0}{P} \right)^{\frac{R_d}{C_p}}, \tag{3}$$

with $R_d/C_p = 0.28571$, the air temperature ($T$), the local surface pressure ($P_0$), and the local air pressure ($P$). Multiple barometers were employed in Perdigão-2017, but none on the 100 m masts, hence, we obtained the local air pressure from the nearest towers with the highest data availability. The selected towers contained measurements only at 2 m a.g.l ($P_2$), which we took as $P_0$.

The pressure $P$ in the 100 m tower was calculated using the barometric formula (Lente and Ősz, 2020):

$$P = P_2 \left( 1 - \frac{\Gamma \Delta z_{asl}}{T_2} \right)^{\frac{gM}{R\Gamma}}, \tag{4}$$

where $T_2$ is the air temperature at 2 m a.g.l., $\Gamma = 0.0065\,\mathrm{K\,m^{-1}}$ is the standard atmosphere lapse rate (Stull, 2017), $M = 0.028\,964\,4\,\mathrm{kg\,mol^{-1}}$ is the average molar mass of Earth's air, and $R = 8.314\,459\,8\,\mathrm{J\,mol^{-1}\,K^{-1}}$ is the universal gas constant. $\Delta z_{asl} = z_{asl,t} - z_{asl,b}$ m accounts for terrain elevation differences between the a.s.l. heights of the temperature/humidity sensor ($z_{asl,t}$) and the nearest selected barometer ($z_{asl,b}$).

Since the humidity sensors on the 100-m towers measured the relative humidity of the air ($RH$), we calculated $r$ using the vapour pressure ($e$) and the air pressure ($P$) (Stull, 2017):

$$r = 0.622 \frac{e}{P - e}. \tag{5}$$

The vapour pressure was calculated by:

$$e = \frac{RH e_s}{100}, \tag{6}$$

where $e_s$ is the saturated vapour pressure, given by:

$$e_s = e_0 \exp \left[ \frac{L}{R_v} \left( \frac{1}{T_0} - \frac{1}{T} \right) \right], \tag{7}$$

with $e_0 = 0.6113 \times 10^3$ Pa, $L = 2.5 \times 10^6$ J kg$^{-1}$, $R_v = 461$ J kg$^{-1}$ K$^{-1}$, and $T_0 = 273.15$ K.

Considering valid measurement periods, the differences between the 10 min averaged $\Theta_v$ and $\Theta$ at the three 100 m towers did not exceed 3.8 K at 100 m a.g.l. during the entire IOP. This resulted in similar $Ri_B$ values when calculated with either $\Theta$ or $\Theta_v$, and changes in the stability classification for only a minority of periods. However, the availability of barometric measurements reduced the number of periods for which we could calculate $Ri_B$ and classify atmospheric stability (e.g., from 99.1 % with $\Theta$ to 64.3 % with $\Theta_v$ in tse09/T25). Consequently, we decided to retain the $Ri_B$ calculation assuming relatively dry air conditions, using $\Theta$ instead of $\Theta_v$. For $\Theta$, we used the approximation $\Theta \approx T + (g/C_p)z$ (Stull, 1988), which nevertheless showed maximum differences of about $3 \times 10^{-2}$ K compared to the formulation in Equation 3.

In the manuscript, we have clarified this choice (line 344) and the small impact of assuming relatively dry air conditions in the paragraph starting at line 351.

CHANGES:

line 344: " The bulk Richardson number ($Ri_B$) was calculated with the 10 min average horizontal mean wind speed components measured at 100 m a.g.l. ($u_{100}$ and $v_{100}$) and assuming relatively dry air conditions, i.e., using the 10 min average potential temperature at 2 m ($\Theta_2$) and 100 m ($\Theta_{100}$) height rather than the virtual potential temperature ..."

line 351: "We assumed relatively dry air conditions ($\Theta_v \approx \Theta$) due to the lack of pressure measurements on Perdigão's 100 m towers and the limited availability of barometric data from nearby towers, which reduced the number of periods for which we could calculate $Ri_B$ and classify atmospheric stability. This assumption proved valid because the differences between the 10 min average $\Theta_v$ and the 10 min average $\Theta$ at the three 100 m towers did not exceed 3.8 K at 100 m a.g.l. during the entire IOP."

(e) Page 18, line 332: "converting the mean temperature into potential temperature". Why "mean" is used in this sentence (or not twice: mean temperature to mean potential temperature)? The time averaging of the temperature data, and the conversion to (virtual) potential temperature are two separate steps. Only in the next paragraph it becomes clear that with mean temperature probably 10-minutes averaged temperature is meant.

RESPONSE:

Thank you for highlighting this point. We have altered the paragraph starting at line 344.

CHANGES:

line 344: "The bulk Richardson number ($Ri_B$) was calculated with the 10 min average horizontal mean wind speed components measured at 100 m a.g.l. ($u_{100}$ and $v_{100}$) and assuming relatively dry air conditions, i.e., using the 10 min average potential temperature at 2 m ($\Theta_2$) and 100 m ($\Theta_{100}$) height rather than the virtual potential temperature (Stull, 1988):

$$Ri_B = \frac{g(\Theta_{100} - \Theta_2)\Delta z}{\Theta_{100}\left[(u_{100})^2 + (v_{100})^2\right]}.$$

The gravitational acceleration is $g = 9.81\,\mathrm{m\,s}^{-2}$, $\Delta z = (100 - 2)$ m, and the wind speed at 2 m a.g.l. was assumed equal to zero. The 10 min average potential temperature was approximated by $\Theta \approx T + (g/C_p)z$, where $g/C_p = 0.0098\,\mathrm{K\,m}^{-1}$ and $T$ is the 10 min average air temperature (Stull, 1988) measured by the temperature sensors."

**Technical corrections**

(f) Page 11, line 213: "…except for the y-wind component measured by VM1." I guess "y-wind component" is a typo here, because throughout the manuscript u- and v-components are used.

RESPONSE:

We have replaced "y-wind component" to "v-wind component" to maintain consistency (line 224).

CHANGES:

ine 224: "...except for the $v$-wind component measured by VM1."

(g) Color usage in the various figures. The different types of blue is hard to distinguish, which is an issue for Figures 5 and 9.

RESPONSE:

We revised the figures to use more distinguishable colours.

**Responses to comments by Reviewer 2 (Joachim Reuder)**

**NOTE:** Our responses to the Reviewer's comments appear in the 'RESPONSE' grey-shaded text box, and the changes in the manuscript are shown in the 'CHANGES' blue-shaded text box.

The manuscript compares systematically wind speed and turbulence quantities obtained from scanning Doppler wind lidar measurements in virtual mast (VM) mode with corresponding sonic anemometer measurements on co-located meteorological towers. The topic is interesting and highly relevant for a wide range of atmospheric boundary layer applications (e.g. wind energy meteorology) where our present measurement capabilities are limited by the availability and height of existing masts. Proving that lidars could extend our corresponding measurement capabilities will therefore open a wide range of new applications. The topic fits very well in the scope of AMT and I think that the manuscript can be considered for publication after some major revisions.

**General comments**

My two main critics are related to a) the description, handling and interpretation of the vertical velocity component and b) the analysis with respect to atmospheric stability presented in in section 4.1.

(a) It has to be carefully explained how your data have been tilt corrected, because this will strongly influence your results (see also specific comments 7b, 9 and 13). If I understand correctly, you argue that the assumption of 0 average vertical wind speed is backed up by the sonic anemometer measurements on the masts. But if you apply tilt correction to the sonics, that is of course no surprise. Only a wind speed and wind direction dependent analysis of systematic deviations could reveal what portion of the tilt is caused by instrument mis-alignment and what by potential tilt of the streamlines due to the topography. This has to be elaborated in much more detail throughout the manuscript.

> RESPONSE:
>
> The quality controlled High-rate Integrated Surface Flux System (ISFS) surface flux data, in a geographic coordinate system, and tilt corrected is available at UCAR/NCAR - Earth Observing Laboratory (2019a). According to the Data Report (UCAR/NCAR - Earth Observing Laboratory, 2019b), sonic anemometer data were tilt-corrected using DTU multistation measurements (Menke and Mann, 2017) to determine the azimuth, pitch, roll, and height of each anemometer, ensuring that the post-processed wind components were represented in geographical coordinates. The DTU multistation, composed of Leica MultiStation MS50, Leica GS14 GSNN Antenna, Leica CRT16 Bluetooth

Cap, and 360° retroreflector, measured at four points on each of the installed sonic anemometers: two on the boom and two on the instrument (Menke and Mann, 2017).

Therefore, the tilt correction method that relies on the assumption of zero averaged vertical velocity was not employed in the post-processing of this data. This means that we can rely on all measured wind components of the post-processed sonic data and that no correction to the manuscript methodology is necessary. We have added an explanation of the sonic anemometer tilt correction to the manuscript (line 118).

CHANGES:

line 118: "The sonic anemometer data was tilt-corrected using laser survey measurements (Menke and Mann, 2017) to determine the azimuth, pitch, roll, and height of each anemometer, ensuring that the post-processed wind components were represented in geographical coordinates (UCAR/NCAR - Earth Observing Laboratory, 2019b)."

(b) Stability is for sure a parameter to be investigated here, and I see this part of the analysis as the most important and novel investigation of your study, Unfortunately, is your use of two stability classes in my opinion not appropriate for this purpose. I suggest, to re-perform the analysis with at least 3 stability classes including a near-neutral range. In this context it would be very helpful to see a histogram of the Richardson numbers occurring in your analysis (that is a plot I really miss in the study), that then could guide you to a proper selection of the near neutral range. In case you see also a decent number of very stable and very unstable conditions, you could even consider to extend your analysis to five stability classes.

RESPONSE:

In response to the reviewer's suggestion, we have added the $Ri_B$ histogram and its discussion to the manuscript (paragraph starting at line 355). For consistency, the atmospheric stability classification was performed following a previous multi-lidar work in Perdigão by Menke et al. (2019).

Regarding the stability classes, we acknowledge that there are different formulations of the bulk Richardson number and definitions of stability classes based on their values. However, each method presents some degree of uncertainty, and the "correct" way to classify the atmosphere's stability is still an open question, especially in complex terrain. Therefore, we opted to keep the classification into unstable ($Ri_b < 0$) and stable ($Ri_b > 0$), using the $Ri_B$ formulation of Stull (1988).

CHANGES:

line 355: "The distribution of the $Ri_B$ values at the three $100\,\mathrm{m}$ towers (Fig. 9) further highlights the different conditions between ridge and valley wind flow. For tse04/T20 and tse13/T29, the histograms peak around zero $Ri_B$ with nearly symmetrical distributions, showing similar quantities of unstable and stable conditions. The valley tower, on the other hand, has a broader distribution with a significant spread towards positive $Ri_B$ values, indicating greater variability in stability compared to the ridge towers and a prevalence of stable atmospheric conditions."

[Figure]

Figure 9: Histogram of the bulk Richardson number from $10\,\mathrm{min}$ average measurements at tse04/T20 (SW ridge), tse09/T25 (valley), and tse13/T29 (NE ridge) at $100\,\mathrm{m}$ a.g.l. during the VM measurement periods of the mean flow. The $Ri_B$ values are constrained to the -10 to 10 interval, with a bin width of 0.2.

(c) As a last general comment I suggest to rework/rephrase the introduction with respect to structure and non-precise scientific writing (I mentioned a few examples in my specific comments).

RESPONSE:

We improved the introduction in the revised manuscript.

**Specific comments**

1. line 45: dual RHI scanning has recently also been used for the detection and characterization of thermal updrafts in the CBL (Duscha, C., Pálenik, J., Spengler, T., and Reuder, J.: Observing atmospheric convection with dual-scanning lidars, Atmos. Meas. Tech., 16, 5103–5123, https://doi.org/10.5194/amt-16-5103-2023, 2023.); this work also documents the potential of retrieving valid data below a fixed user-defined CNR threshold (comment 9)

> RESPONSE:
>
> Thank you for pointing out this study, we have included its reference in the manuscript (line 53).

> CHANGES:
>
> line 53: "Recent studies include experiments in complex terrain (Hill et al., 2010; Cherukuru et al., 2015; Santos et al., 2020; Duscha et al., 2023) and urban environments (Collier et al., 2005; Newsom et al., 2005; Calhoun et al., 2006; Wittkamp et al., 2021)."

2. line 73: "University of Porto, 2020"; is there a more proper reference, e.g. once again Fernando et al.?

> RESPONSE:
>
> We have changed the reference for Fernando et al. (2019).

> CHANGES:
>
> line 72: "During the campaign, profiler (8) and scanning (18) lidars were deployed (Fernando et al., 2019)."

3. line 73: "were configured with different scanning strategies"; please rephrase, you can't configure a strategy.

> RESPONSE:
>
> Yes, this sentence was altered in the revised manuscript (line 73).

> CHANGES:
>
> line 73: "The latter operated with different scanning schemes, including RHIs along the ridges, across the ridges (in three transects), and coordinated setups forming dual-lidar measurements."

4. line 73/74: "enabling the retrieval of multi-lidar measurements"; non-precise formulation, please rephrase; you use multiple lidar measurements to retrieve some other parameters

> RESPONSE:
>
> Yes, this sentence was altered in the revised manuscript (line 73).

CHANGES:

line 73: "The latter operated with different scanning schemes, including RHIs along the ridges, across the ridges (in three transects), and coordinated setups forming dual-lidar measurements."

5. line 90: replace "on" by "in"

RESPONSE:

Thank you. This word was altered in the revised manuscript (line 90).

CHANGES:

ine 90: " With wind turbines increasingly being placed in complex terrains ..."

6. naming of the towers/virtual masts (table 1 and throughout the whole text): Do you really need the complicated double numbering/labeling; it would be much easier readable if you would go for one clear and understandable abbreviation. My suggestion WS2, WS3, ... for the WindScanners, and maybe T1, T2, T3 for the towers, that would then nicely coincide with the corresponding virtual masts VM1/2/3? As it is it is really complicated to read and requires continuous look up again.

RESPONSE:

The employed tower/WindScanner/VM names are the original names used in the Perdigão experiment, which allows for direct comparison with other works made in Perdigão.

7. line 115: can you elaborate a bit more on the pre-processing;

   a) which criteria was used for spike detection?

   b) what exactly do you mean with tilt correction (Planar Fit?, Double-rotation?, Triple-rotation?). This will have an important influence on the interpretation of the data afterwards.

RESPONSE:

The High-rate Integrated Surface Flux System (ISFS) surface flux data we used was already pre-processed by UCAR/NCAR, available at UCAR/NCAR - Earth Observing Laboratory (2019a), which is quality controlled, in geographic coordinates, and tilt corrected.

(a) The spiking detection of this pre-processed data employed the methodology

from Hojstrup (1993), which is detailed in UCAR/NCAR - Earth Observing Laboratory (2024). In this procedure, a data point $(x_i)$ is identified as a spike if it deviates from a forecasted point $(x_f)$ by more than a discrimination level $(L)$ times the standard deviation $(\sigma_i)$: $|x_i - x_f| > L\sigma_i$.

Running statistics are used to calculate the mean $(m_i)$, auto correlation $(c_i)$, and variance $(\nu_i)$ of the $i$-th data point. Then, the forecasted point is computed by: $x_f = x_{i-1}c_i + (1 - c_i)m_i$.

The initial discrimination level $L$ is based on the minimum probability of a spike, typically $1 \times 10^{-5}$, and adjusted by a level factor, usually 2.5. This discrimination level is periodically updated, every 25 points, based on the auto-correlation of the data.

(b) As mentioned, sonic anemometer data were tilt-corrected by UCAR/NCAR using DTU multistation measurements of azimuth, pitch, roll, and height for each anemometer (Menke and Mann, 2017), ensuring that the post-processed wind components were represented in geographical coordinates (UCAR/NCAR - Earth Observing Laboratory, 2019b).

8. line127-128: I feel that -22dB is a very conservative threshold, can you elaborate on the amount of data you are losing by applying this threshold;

RESPONSE:

The threshold value of $-22\,\mathrm{dB}$ was determined based on CNR vs. radial velocity plots from the multiple WindScanners. The filter was applied before the dual-lidar processing. While this does result in some data loss, improving the data/noise filtering lied beyond the scope of the article.

CHANGES:

line 131: "...the WS data were initially filtered out according to the equipment's radial velocity limits ($[-30, 30]$ m s$^{-1}$) and the carrier-to-noise ratio (CNR), where a threshold equal to $-22\,\mathrm{dB}$ (determined from CNR versus radial velocity plots of the multiple WindScanners) was imposed."

9. line 145: "... assuming the vertical wind component is zero (w = 0)" ; how confident are you that this assumption holds in the complex environment of Perdigao? (see also my comments 7b and 13)

RESPONSE:

Based on the sonic anemometer measurements at approximately $100\,\mathrm{m}$ a.g.l., the $10\,\mathrm{min}$ average vertical velocity did not exceed $3.6\,\mathrm{m\,s}^{-1}$ during the entire

IOP. Specifically, the 10 min average vertical velocity was $0 \pm 0.5$ m s$^{-1}$ around 59 % of the IOP period at tse04/T20, 82 % at tse09/T25, and 70 % at tse13/T29 (Fig. 1). Consequently, we consider the assumption of zero vertical velocity to be valid for retrieving the wind components from dual-lidar measurements at 80 and 100 m a.g.l. in Perdigão.

CHANGES:

line 392: "This minimal impact is attributed to the small elevation angles of the lidars' beams (Table 2) and the low vertical velocity at the site, which did not exceed $3.6$ m s$^{-1}$ at 100 m a.g.l. during the IOP."

[Figure]

Figure 1: Histogram of the 10 min average vertical wind velocity at tse04/T20 (SW ridge), tse09/T25 (valley), and tse13/T29 (NE ridge) measurements at 100 m a.g.l. during the intensive observational period. $N$ represents the total number of valid 10 min average measurements from the sonic anemometers at 100 m a.g.l. during the IOP.

10. line 198/199; " is the radial velocity error, assuming that is identical in both lidars"; Do you also assume that the error is constant along the beam?; my experience with the scanning WindCube systems is that they have an individual "focus" area where they are performing better, which could cause both distance dependent variations in the errors, as well as differences between the different lidars. This could have considerable implications on your error estimates. Maybe you can elaborate a bit more on that, I assume that DTU has quite good control on their deployed lidars with respect to this behavior.

RESPONSE:

It is correct that the error is assumed identical for all lidars and independent of distance. We acknowledge that line-of-sight error depends on the individual lidar, the amount of backscatter in the atmosphere, the distance from the instrument, the focus position, and the instrument's temperature. However,

a larger contributor to the error is the angle between the beams, and that is the focus of this discussion. We have added a sentence clarifying that in the manuscript (line 208).

CHANGES:

line 208: "While the radial velocity error depends on several factors, such as the specific lidar, atmospheric backscatter, distance from the instrument, focus position, and instrument temperature, we assume it to be identical in both lidars because the angle between the beams is a more significant contributor to the dual-lidar error."

11. line 211: "assuming that u and v are not correlated"; aren't u and v closely correlated by the wind direction?

RESPONSE:

For the dual-lidar error propagation of the horizontal wind speed, we assumed that the errors in $u$ and $v$ are uncorrelated ($\sigma_{uv} = 0$). This assumption is valid for an atmospheric boundary layer when ignoring the Coriolis force. However, we acknowledge that in a real flow over complex terrain, this assumption may be more questionable. In the manuscript, we opted for this assumption since it was used solely for an error estimate.

We have clarified this in the manuscript (line 222).

CHANGES:

line 222: " ...assuming that the errors in $u$ and $v$ are not correlated."

12. line 226: replace ")" by "]"

RESPONSE:

We used ")" to represent that it is an open (and not closed) interval.

13. subchapter 4.2.2 Vertical velocity (lines 359-363): What kind of are you using for the vertical velocity (see also my comment on that before in section 2.2 describing the tower data)? This could distinctly influence your results as the different tilt correction methods (that are basically designed to bring the vertical wind speed on average to zero) would cover potential systematic vertical velocities, e.g. caused by the terrain. For that it would be helpful to look into the non-corrected raw data and a potential systematic wind direction and wind speed dependent bias in the vertical velocities.

RESPONSE:

As mentioned in responses (a) and 7.b), the data were tilt-corrected using, solely, local geometrical measures to ensure the alignment between the wind components' referential and the geographical coordinates. No assumption of zero average vertical velocity was made; hence, there should be no risk of attenuating vertical velocities, or masking of the terrain or thermal effects. For this reason, we believe that it should not be necessary to evaluate biases between the pre- and post-tilt-correction data.

14. line 365 "progressively lower sampling rates": How did you lower the sampling rate, by just picking e.g. every 10th value or averaging over the ten corresponding values and using the mean for further analysis?

RESPONSE:

The original 20 Hz wind-component data arrays were structured as [time, sample], with time in seconds and 20 samples per second. For frequencies in the $[1, 20)$ Hz interval, we down-sampled the data by selecting every $n$-th sample from the original dataset (e.g., for a frequency of 2 Hz from the 20 Hz dataset, every 10th sample was selected, as $u_{2\,\text{Hz}} = u[:,0::10]$). For frequencies below 1 Hz, we selected the $n$-th time step from the original dataset and the first sample (e.g., for a frequency of 0.5 Hz from the 20 Hz dataset, every 2nd time step was selected, as $u_{0.5\,\text{Hz}} = u[0::2,0]$). After down-sampling, we calculated the variances and averages over 10 min intervals.
An explanation of the procedure was added in the manuscript (line 401).

CHANGES:

line 401: "The data were down-sampled by selecting every $n$-th sample for frequencies between 1 Hz and 20 Hz (e.g., for 2 Hz, every 10th sample), and by selecting the $n$-th time step for frequencies below 1 Hz (e.g., for 0.5 Hz, every 2nd time step). Following down-sampling, variances and averages were calculated over 10 min intervals. "

15. figure 9 and corresponding text lines 369-374: wouldn't it be much more straightforward/"honest" to present this (at least for the velocity) for the horizontal velocity instead of only one component to avoid any potential wind direction influence?

RESPONSE:

We wanted to represent the influence of the sampling rate on the $RMSE$ for both mean and turbulent variables. As shown in Fig. 2, the sampling rate similarly influenced the $RMSE$ of the $u$- and $v$-wind components of the same moment, and for the mean flow, the $RMSE$ of $V_h$ exhibited results

comparable to those of $u$ and $v$. Therefore, we chose to present the graph for a single mean and turbulent wind component.

Nonetheless, to address potential concerns regarding wind direction influence, we have included the averaged statistical metrics for the horizontal wind speed due to sampling rate in Table 9, averaged for the three masts at 100 m a.g.l., and in lines 412 and 416.

CHANGES:

line 412: "At 100 m a.g.l., the estimated average $RMSE$ of the VMs, due solely to their sampling rate, ranged between 0.102 and 0.180 m s$^{-1}$ for the mean flow quantities ... (Table 9)."

line 416: "For the mean horizontal wind velocity, 33 % of the VMs' average $RMSE$ at 100 m a.g.l. can be attributed to their measurement frequency."

[Figure]

(a)        (b)        (c)

Figure 2: $RMSE$ of sonic measurements by the sampling rate, for the mean ($u$ and $v$) and turbulent ($u'u'$ and $v'v'$) wind speed components and for the mean wind velocity ($V_h$), on the three 100 m towers at 100 m a.g.l. The $RMSE$ units are [m s$^{-1}$] for $u$, $v$, and $V_h$ and [m$^2$ s$^{-2}$] for $u'u'$ and $v'v'$.

Table 9: Averaged statistical metrics due to sampling rates in the virtual-mast measurement range for the mean (0.018–0.038 Hz) and turbulent (0.030–0.038 Hz) flow, based on sonic readings at 100 m a.g.l.

| Metric | Mean flow | | | Turbulent flow | |
|---|---|---|---|---|---|
| | $u$ | $v$ | $V_h$ | $u'u'$ | $v'v'$ |
| $r$ | 0.995–0.998 | 0.996–0.999 | 0.992–0.997 | 0.911–0.931 | 0.930–0.945 |
| $RMSE$ | 0.104–0.180 | 0.104–0.179 | 0.102–0.178 | 0.262 – 0.300 | 0.267–0.306 |
| $Bias$ | 0.001–0.002 | $\sim$0––0.001 | 0.003–0.008 | $-0.012$––0.016 | $-0.011$––0.015 |

16. the references Menke et al., (line 498) and Pitter et al. (line 516) seem to be incomplete

> RESPONSE:
>
> Menke et al. (2018) (dataset) and Pitter et al. (2012) (conference paper) references were corrected.

**References**

Calhoun, R., Heap, R., Princevac, M., Newsom, R., Fernando, H., and Ligon, D.: Virtual towers using coherent Doppler lidar during the Joint Urban 2003 Dispersion Experiment, Journal of Applied Meteorology and Climatology, 45, 1116–1126, URL http://www.jstor.org/stable/26171770, 2006.

Cherukuru, N. W., Calhoun, R., Lehner, M., Hoch, S. W., and Whiteman, C. D.: Instrument configuration for dual-Doppler lidar coplanar scans: METCRAX II, Journal of Applied Remote Sensing, 9, 096 090, https://doi.org/10.1117/1.JRS.9.096090, 2015.

Collier, C. G., Davies, F., Bozier, K. E., Holt, A. R., Middleton, D. R., Pearson, G. N., Siemen, S., Willetts, D. V., Upton, G. J. G., and Young, R. I.: Dual-Doppler Lidar Measurements for Improving Dispersion Models, Bulletin of the American Meteorological Society, 86, 825–838, https://doi.org/10.1175/BAMS-86-6-825, 2005.

Duscha, C., Pálenik, ., Spengler, T., and Reuder, J.: Observing atmospheric convection with dual-scanning lidars, Atmospheric Measurement Techniques, 16, 5103–5123, https://doi.org/10.5194/amt-16-5103-2023, 2023.

Fernando, H. J. S., Mann, J., Palma, J. M. L. M., Lundquist, J. K., Barthelmie, R. J., Belo-Pereira, M., Brown, W. O. J., Chow, F. K., Gerz, T., Hocut, C. M., Klein, P. M., Leo, L. S., Matos, J. C., Oncley, S. P., Pryor, S. C., Bariteau, L., Bell, T. M., Bodini, N., Carney, M. B., Courtney, M. S., Creegan, E. D., Dimitrova, R., Gomes, S., Hagen, M., Hyde, J. O., Kigle, S., Krishnamurthy, R., Lopes, J. C., Mazzaro, L., Neher, J. M. T., Menke, R., Murphy, P., Oswald, L., Otarola-Bustos, S., Pattantyus, A. K., Rodrigues, C. V., Schady, A., Sirin, N., Spuler, S., Svensson, E., Tomaszewski, J., Turner, D. D., van Veen, L., Vasiljević, N., Vassallo, D., Voss, S., Wildmann, N., and Wang, Y.: The Perdigão: peering into microscale details of mountain winds, Bulletin of the American Meteorological Society, 100, 799–819, https://doi.org/10.1175/BAMS-D-17-0227.1, 2019.

Hill, M., Calhoun, R., Fernando, H. J. S., Wieser, A., Dörnbrack, A., Weissmann, M., Mayr, G., and Newsom, R.: Coplanar Doppler lidar retrieval of rotors from T-REX, Journal of the Atmospheric Sciences, 67, 713–729, https://doi.org/10.1175/2009JAS3016.1, 2010.

Hojstrup, J.: A statistical data screening procedure, Measurement Science and Technology, 4, 153–157, https://doi.org/10.1088/0957-0233/4/2/003, 1993.

Lente, G. and Ősz, K.: Barometric formulas: various derivations and comparisons to environmentally relevant observations, ChemTexts, 6, 13, https://doi.org/10.1007/s40828-020-0111-6, 2020.

Menke, R. and Mann, J.: Perdigao 2017: Laser survey of measurement masts, Tech. rep., DTU Wind Energy, URL https://perdigao.fe.up.pt/documents/file/238, 2017.

Menke, R., Mann, J., and Vasiljevic, N.: Perdigão-2017: multi-lidar flow mapping over the complex terrain site, Technical University of Denmark, https://doi.org/10.11583/DTU.7228544.V1, 2018.

Menke, R., Vasiljević, N., Mann, J., and Lundquist, J. K.: Characterization of flow recirculation zones at the Perdigão site using multi-lidar measurements, Atmospheric Chemistry and Physics, 19, 2713–2723, https://doi.org/10.5194/acp-19-2713-2019, 2019.

Newsom, R. K., Ligon, D., Calhoun, R., Heap, R., Cregan, E., and Princevac, M.: Retrieval of microscale wind and temperature fields from single- and dual-Doppler lidar data, Journal of Applied Meteorology, 44, 1324–1345, https://doi.org/10.1175/JAM2280.1, 2005.

Pitter, M., Abiven, C., Vogstad, K., Harris, M., Barker, W., and Brady, O.: Lidar and computational fluid dynamics for resource assessment in complex terrain, in: Proceedings EWEA, Copenhagen, Denmark, 2012.

Santos, P., Mann, J., Vasiljević, N., Cantero, E., Sanz Rodrigo, J., Borbón, F., Martínez-Villagrasa, D., Martí, B., and Cuxart, J.: The Alaiz experiment: untangling multi-scale stratified flows over complex terrain, Wind Energy Science, 5, 1793–1810, https://doi.org/10.5194/wes-5-1793-2020, 2020.

Stull, R. B.: An introduction to boundary layer meteorology, Springer Netherlands, Dordrecht, https://doi.org/10.1007/978-94-009-3027-8, 1988.

Stull, R. B.: Practical meteorology: an algebra-based survey of atmospheric science, Dept. of Earth, Ocean & Atmospheric Sciences, University of British Columbia, Vancouver, BC, Canada, version 1.02b edn., 2017.

UCAR/NCAR - Earth Observing Laboratory: NCAR/EOL Quality Controlled High-rate ISFS surface flux data, geographic coordinate, tilt corrected. Version 1.1, https://doi.org/10.26023/8X1N-TCT4-P50X, 2019a.

UCAR/NCAR - Earth Observing Laboratory: Perdigão-ISFS Data Report, URL https://www.eol.ucar.edu/content/perdigao-isfs-data-report, 2019b.

UCAR/NCAR - Earth Observing Laboratory: Despiking algorithm, URL https://www.eol.ucar.edu/content/despiking-algorithm, 2024.

Wittkamp, N., Adler, B., Kalthoff, N., and Kiseleva, O.: Mesoscale wind patterns over the complex urban terrain around Stuttgart investigated with dual-Doppler lidar profiles, Meteorologische Zeitschrift, 30, 185–200, https://doi.org/10.1127/metz/2020/1029, 2021.